# Anthropogenic and biogenic influence on VOC fluxes at an urban background site in Helsinki, Finland

Pekka Rantala[1], Leena Järvi[1], Risto Taipale[1], Terhi K. Laurila[1], Johanna Patokoski[1], Maija K. Kajos[1], Mona Kurppa[1], Sami Haapanala[1], Erkki Siivola[1], Tuukka Petäjä[1], Taina M. Ruuskanen[1], and Janne Rinne[1,2,3,4]

[1]Department of Physics, University of Helsinki, Helsinki, Finland
[2]Department of Geoscience and Geography, University of Helsinki, Helsinki, Finland
[3]Finnish Meteorological Institute, Helsinki, Finland
[4]Department of Physical Geography and Ecosystems Science, Lund University, Lund, Sweden

*Correspondence to:* Pekka Rantala (pekka.a.rantala@helsinki.fi)

**Abstract.** We measured volatile organic compounds (VOC), carbon dioxide ($CO_2$) and carbon monoxide (CO) at an urban background site near the city centre of Helsinki, Finland, Northern Europe. The VOC and $CO_2$ measurements were obtained between January 2013 and September 2014 whereas for CO a shorter measurement campaign in April–May 2014 was conducted. Both anthropogenic and biogenic sources were identified for VOCs in the study. Strong correlations between VOC fluxes and CO fluxes and traffic rates indicated anthropogenic source of many VOCs. The VOC with the highest emission rate to the atmosphere was methanol which originated mostly from traffic and other anthropogenic sources. The traffic was also a major source for aromatic compounds in all seasons whereas isoprene was mostly emitted from biogenic sources during summer. Some amount of traffic related isoprene emissions were detected during other seasons but this might have also been an instrumental contamination from cycloalkane products. Generally, the observed VOC fluxes were found to be small in comparison with previous urban VOC flux studies. However, the differences were probably caused by lower anthropogenic activities as the $CO_2$ fluxes were also relatively small at the site.

## 1 Introduction

Micrometeorological flux measurements of volatile organic compounds (VOC) in urban and semi-urban areas are limited, although local emissions have a major effect on the local and regional atmospheric chemistry and furthermore on air quality (e.g. Reimann and Lewis, 2007 and references therein). Biogenic VOCs, mainly isoprene and monoterpenes, affect hydroxyl radical (OH) concentration and aerosol particle growth (Atkinson, 2000; Atkinson and Arey, 2003; Kulmala et al., 2004; Spracklen et al., 2008; Kazil et al., 2010; Paasonen et al., 2013). Long-lived compounds, such as anthropogenically emitted benzene, contribute also to the VOC concentrations in rural areas (e.g. Patokoski et al., 2014, 2015).

The VOCs may have both anthropogenic and biogenic sources in the urban areas which complicates the analysis of VOC flux measurements made in these areas. Globally, the most important anthropogenic sources are traffic, industry, gasoline evaporation and solvent use (Watson et al., 2001; Reimann and Lewis, 2007; Kansal, 2009; Langford et al., 2009; Borbon et al., 2013

and references therein) whereas the biogenic VOC sources within cities include mostly urban vegetation, such as trees and shrubs in public parks and in street canyons. Based on previous micrometeorological flux studies, the urban areas are observed to be a source for methanol, acetonitrile, acetaldehyde, acetone, isoprene+cycloalkanes, benzene, toluene and $C_2$-benzenes (Velasco et al., 2005; Langford et al., 2009; Velasco et al., 2009; Langford et al., 2010; Park et al., 2010; Valach et al., 2015).

In addition, concentration measurements connected to source models have underlined emissions of various other VOCs, such as light hydrocarbons, from the urban sources (e.g. Watson et al., 2001; Hellén et al., 2003, 2006, 2012). Monoterpene emissions have surprisingly remained mainly unstudied, although the monoterpenes have generally major effects on atmospheric chemistry. For example, Hellén et al. (2012) found that monoterpenes and isoprene together have a considerable role in OH-reactivity in Helsinki, Southern Finland. The biogenic emissions might have also a considerable role in ozone ($O_3$) chemistry in the urban areas (e.g. Calfapietra et al., 2013).

The VOC flux measurements reported in literature have been conducted in the latitudes ranging from 19°N to 53°N, but most of the measurement in the north have been conducted in the UK where winters are relatively mild (Langford et al., 2009, 2010; Valach et al., 2015). Thus no measurements have been reported from the northern continental urban areas. The VOC emissions from traffic are typically due to incomplete combustion. This also results in emissions of carbon monoxide (CO), and thus the emissions of certain VOCs are potentially linked with CO fluxes. However, only two publications on the urban VOC fluxes combine the VOC fluxes with the CO fluxes in their analysis (Langford et al., 2010; Harrison et al., 2012). Thus our aim is i) to characterize the VOC fluxes in a northern urban city over an annual cycle, ii) to identify the main sources, such as traffic and vegetation, of aromatics, oxygenated VOCs and terpenoids taking into account traffic volume together with the measured CO and carbon dioxide ($CO_2$) fluxes and the ambient temperature ($T$), and iii) to compare the VOC fluxes with the previous urban VOC flux studies to assess the relation of the VOC fluxes and the CO and $CO_2$ fluxes in different cities.

## 2 Materials and methods

### 2.1 Measurement site and instrumentation

Measurements were carried out at urban background station SMEAR III in Helsinki (60° 12' N, 24° 58' E, Järvi et al., 2009a). The population of Helsinki is around 630 000 (http://vrk.fi/default.aspx?docid=8882&site=3&id=0, cited in 12 December 2015). The site is classified as a local climate zone, which corresponds to "an open low-rise" category (see Stewart and Oke, 2012) with detached buildings and scattered trees and abundant vegetation. The site is in a humid continental climate zone with a clear annual variation between the four seasons: monthly mean temperature varies from $-4.9°C$ in February to $17.6°C$ in July (1971–2000, Drebs et al., 2002; see also Fig. 1), and daylight hours range from 6 to 19 h per day. The SMEAR III site consists of a 31-m-tall lattice tower located on a hill, 26 m above the sea level and 19–21 m above the surrounding terrain. The site is roughly five kilometres North-East from Helsinki City Centre. According to local wind direction, the surroundings around the tower can be divided into three areas: built, road and vegetation (Vesala et al., 2008, Table 1, Fig. 2). However, a fraction of vegetation was significant also in the built and road sectors (Table 1). Thus, a better name for the built sector

would be, for example, "urban residential sector with vegetation", but the short names are used throughout the text to keep the terminology and subsequent discussion as simple and short as possible.

The built sector in the northern direction (320°–40°) is dominated by university campus buildings and Finnish Meteorological Institute (mean height 20 m) close to the tower. In the road sector (40°–180°), one of the main roads leading to Helsinki city centre passes through with the closest distance between the road and the tower being 150 m. The area in-between is covered by deciduous forest with mainly birch (*Betula sp.*), Norway maple (*Acer platanoides*), aspen (*Populus tremula*), goat willow (*Salix caprea*) and bird cherry (*Prunus padus*) (Vesala et al., 2008, Fig. 2). On the road, a typical workday traffic rate is around 44 000 vehicles per day (Lilleberg and Hellman, 2011), and the vehicles have been found to be the main source of $CO_2$ and aerosol particle number emissions in the area (Järvi et al., 2012; Ripamonti et al., 2013). In the vegetation sector (180°–320°), most of the surface is covered by green areas of the Kumpula Botanic Garden and the City Allotment Garden. During this study, the wind blew most often from the vegetation sector and least from the built sector.

The site infrastructure, the flux measurement conditions and the surrounding areas are described in detail in Vesala et al. (2008) and in Järvi et al. (2009a).

### 2.1.1 VOC measurements with PTR-MS and volume mixing ratio calculations

A proton-transfer-reaction quadrupole mass spectrometer (PTR-MS, Ionicon Analytik GmbH, Innsbruck, Austria; Lindinger et al., 1998) was measuring 12 different mass-to-charge ratios (*m/z*, see Table 2) every second hour using a 0.5 s sampling time between 1 January 2013 and 27 June 2014. The total sampling cycle was around 7 s (Fig. 1). For the rest of the time the PTR-MS sampled a wider range of mass-to-charge ratios but those measurements are not considered in this study. In addition, we had a short campaign between 27 June and 30 September 2014 when 14 mass-to-charge ratios were measured using the same 0.5 s sampling time. During the campaign, the two additional mass-to-charge ratios were *m/z* 89 and *m/z* 103. In that period, the measurement cycle took always two hours so that *m/z* 31–69 were measured during the first and *m/z* 79–137 during the second hour, and the total sampling cycle was around 4.5 s. In summer 2014, there were some data gaps due to software problems (Table 2).

The PTR-MS was located inside a measurement container and sample air was drawn to the instrument using a PTFE tubing with 8 mm inner diameter (i.d.). The sample line was 40-m-long and it was heated (10 W m$^{-1}$) to avoid condensation of water vapour. A continuous air-flow was maintained in the tube with some variations in the flow rate: first 20 l min$^{-1}$ (whole year 2013), then 40 l min$^{-1}$ (until 30 May 2014) and then 20 l min$^{-1}$ (until the end of the measurements) again. The corresponding Reynolds numbers were around 3500 and 7000 for the lower and higher sample line flows, respectively. From the main inlet, a side flow of 50–100 ml min$^{-1}$ was drawn to PTR-MS via a 0.5-m-long PTFE tube with 1.6 mm i.d.

The PTR-MS was maintained at a drift tube pressure of 2.0–2.2 mbar and primary ion ($H_3O^+$) count rate of about 10–30·10$^6$ counts per second (cps, measured at *m/z* 21). With these settings, $E/N$-ratio where $E$ is the electric field and $N$ the number density of the gas in the drift tube, was typically around 135 Td (Td $= 10^{-21}$ V m$^2$). Oxygen level $O_2^+$ was mostly below 2% of the $H_3O^+$ signal.

The instrument was calibrated every second or third week using a diluted VOC standard (Apel-Riemer, accuracy $\pm5\%$; Table 2). The volume mixing ratios were calculated using a procedure described in detail in Taipale et al. (2008). Before the calibration, SEM voltage (MasCom MC-217) of the PTR-MS was always optimized to get a high enough primary ion signal level (e.g. Kajos et al., 2015). The optimized SEM voltage was also used in the measurements until the next calibration. The

instrumental background was determined every second hour by sampling VOC free air, produced with a zero air generator (Parker Balzon HPZA-3500-220). The intake for the zero air generator was outside of the measurement cabin close to the ground, thus, the relative humidity was the same for both the zero air measurements and the ambient measurements. During the measurement period, the zero air generator was working sometimes improperly leading to contaminated $m/z$ 93 signal. These periods were removed from the zero air measurements and replaced by the nearest reliable values. In addition, due to

software problems, the zero air measurements were not recorded between 7 July and 30 September 2014. These gaps were replaced by a median diurnal cycle values of the zero air measured during 27 June – 7 July 2014. One should note that the mentioned problems with the zero air measurements had no effect on the flux calculations. However, they did, of course, cause additional uncertainties in the measured concentration levels but a systematic error for the concentration levels was estimated to be negligible.

**2.1.2   Ancillary measurements and data processing**

An ultrasonic anemometer (Metek USA-1, Metek GmbH, Germany) was installed at 31 m, 0.13 m above the VOC sampling inlet. The ambient temperature was also measured at the VOC sampling level with a Pt-100 sensor. Photosynthetic photon flux density was measured at 31 m in the measurement tower using a photodiode sensor (Kipp&Zonen, Delft, Netherlands). Pressure was measured with Vaisala HMP243 barometer on the roof of the University building near the site.

Hourly traffic rates were measured 4 km from the measurement site by the City of Helsinki Planning Department. These rates were converted to correspond to the traffic rates of the road next to the measurement site following the procedure presented in Järvi et al. (2012).

$CO_2$ and CO concentrations (10 Hz) were measured with a Li-Cor 7000 (LI-COR, Lincoln, Nebraska, USA) and a CO/$N_2$O analyser (Los Gatos Research, model N2O/CO-23d, Mountain View, CA, USA; later referred as LGR), respectively. The $CO_2$

concentration was measured continuously between January 2013 and September 2014. The CO concentration was measured between 3 April and 27 May 2014 (Fig. 1) and the LGR was connected to the same main inlet line with the PTR-MS. During the CO measurements, the main inlet flow was 40 l min$^{-1}$. After the LGR was removed from the setup, the main inlet flow was decreased to 20 l min$^{-1}$ to increase the pressure in the sampling tube and to get a higher side flow to the PTR-MS (from 50 to 100 ml min$^{-1}$).

Thirty minute average CO and $CO_2$ fluxes were calculated using the eddy covariance technique from raw data according to commonly accepted procedures (Aubinet et al., 2012). A two-dimensional (2D) coordinate rotation was applied to the wind data and all data were linearly de-trended. The 2D rotation was used instead of a planar fitting as the 2D rotation is likely to be less prone to systematic errors above a complex urban terrain (Nordbo et al., 2012b). Spike removal was made based on a difference limit (Mammarella et al., 2016). Time lags between wind and scalar data were obtained by maximizing the cross-

covariance function. For the CO and $CO_2$ measurements, mean time lags of 5.8 s and 7.0 s, respectively, were obtained. Finally, spectral corrections were applied. The low frequency losses for the fluxes were corrected based on theoretical corrections (Rannik and Vesala, 1999), whereas the high-frequency losses were experimentally determined. Finally, the 30-min fluxes were quality checked for stationarity with a limit of 0.3 (Foken and Wichura, 1996), and the periods with $u_* < 0.2$ m s$^{-1}$ were removed from further analysis. More details of the data post-processing can be found in Nordbo et al. (2012b). The corresponding data coverages for CO and $CO_2$ fluxes were 54.0% and 61.9%, respectively. The random error and detection limit of CO flux were 0.23 $\mu$g m$^{-2}$s$^{-1}$ and 0.16 $\mu$g m$^{-2}$s$^{-1}$, respectively. The corresponding numbers for the $CO_2$ flux were 0.05 $\mu$g m$^{-2}$s$^{-1}$ and 0.03 $\mu$g m$^{-2}$s$^{-1}$, respectively.

## 2.2 VOC flux calculations

### 2.2.1 Disjunct eddy covariance method

In a disjunct eddy covariance method (hereafter DEC), the flux is calculated using a discretized covariance:

$$\overline{w'c'} \approx \frac{1}{n} \sum_{i=1}^{n} w'(i - \lambda/\Delta t)c'(i), \tag{1}$$

where $n$ is the number of measurements during the flux averaging time, $\Delta t$ is a sampling interval and $\lambda$ is a lag time caused by sampling tubes (e.g. Rinne et al., 2001; Karl et al., 2002; Rinne and Ammann, 2012). The VOC fluxes were calculated for each 45-min-period according to Eq. (1) using 385 data points (600 data points between 26 June and 30 September 2016). Before the calculations, the linear trend was removed from the concentration and wind measurements. In addition, the 2D rotation was applied to the wind vectors.

The PTR-MS and the wind data were recorded to separate computers, thus, lag times were shifting artificially as the computer clocks performed unequally. Therefore, we first determined lag times of *m/z* 37 (first water cluster, $H_3O^+H_2O$) for each data set between two calibrations. Then, a linear trend was removed from the lag times to cancel the artificial shift. After that, the shifted cross covariance functions were summed (as in Park et al., 2013), and an average lag-time was determined for each mass-to-charge ratio from the summed cross covariance functions. Finally, the lag-time for each 45-min-period was determined by using a $\pm 2.5$ s lag time window around the previously determined mean lag-time with a smoothed maximum covariance method described in Taipale et al. (2010). The smoothed cross covariance functions were calculated using a running mean with an averaging period of $\pm 2.4$ s. However, if the mean lag-time value was not found, the previous reliable mean lag-time value was used instead. We defined that the mean lag-time was representative if a peak value of the summed cross covariance function was higher than $3\sigma_{tail}$ where $\sigma_{tail}$ is mean standard deviation of the summed cross covariance function tails. The standard deviations were calculated using a lag-time window of $\pm(180 - 200)$ s. If a certain mass-to-charge ratio showed no representative peak values during the whole period at all, its flux values were defined to be insignificant and the mass-to-charge ratio was disregarded from further study.

The lag times were allowed to vary slightly ($\pm 2.5$ s) around the mean lag-times because removing the linear trend potentially caused uncertainties. Moreover, changes in relative humidity might have led to changes in the lag times at least in the case of

methanol which is a water-soluble compound, even with heated inlet line. Thus, the fluxes could be underestimated if the constant lag-times were used (see supplementary material). However, the lag time window we used, was quite narrow, $\pm 2.5$ s, to limit uncertainties ("mirroring effect") caused by the maximum covariance method connected to the fluxes near the detection limit (Langford et al., 2015). Also, one should note that in our case the maximum covariance was determined from the smoothed cross covariance function which already limits the possible overestimation of the measured DEC fluxes, and thus the mirroring effect (Taipale et al., 2010). Some flux values could be slightly underestimated if the correct lag-time was outside of the $\pm 2.5$ s window. Figure S1 shows a comparison of the fluxes that were calculated using a constant lag-time and the fluxes obtained in this study.

The fluxes measured by the DEC method suffer from same sources of systematic underestimation as the fluxes measured by the EC method, including the high and low frequency losses (e.g. Moore, 1986; Horst, 1997). According to Horst (1997), the high frequency losses, $\alpha_{\text{horst}}$, can be estimated using an equation

$$(\alpha_{\text{horst}})^{-1} = \frac{1}{1 + (2\pi f_m \tau)^\beta},$$
(2)

where $\tau$ is the response time of the system, $f_m = n_m \overline{u}/(z_m - d)$ and $\beta = 7/8$ and $\beta = 1$ in unstable and stable stratification, respectively. In here, $\overline{u}$ is the mean horizontal wind, $z_m$ is the measurement height and $d$ corresponds to the zero displacement height. The parameter $n_m$ has been observed to be constant in the unstable stratification at the site ($n_m = 0.1$), and in the stable stratification ($\zeta > 0$) having the following experimental, stability and wind direction dependent values (Järvi et al., 2009b):

$$n_m = \begin{cases} 0.1(1 + 2.54\zeta^{0.28}), \ d = 13 \text{ m}, & \text{(built)} \\ 0.1(1 + 0.96\zeta^{0.02}), \ d = 8 \text{ m}, & \text{(road)} \\ 0.1(1 + 2.00\zeta^{0.27}), \ d = 6 \text{ m}, & \text{(vegetation)} \end{cases}$$
(3)

where $\zeta$ is the stability parameter.

A constant response time of 1.0 s and Eq. (2) were used for the high-frequency flux corrections. The constant value was estimated based on the previous studies with PTR-MS (Ammann et al., 2006; Rantala et al., 2014; Schallhart et al., 2015) where the response time of the measurement setup was estimated to be around 1 s. However, the response time is probably compound dependent as e.g. methanol might have a dependence on the relative humidity (RH) due to its polarity and water solubility. The response time of water vapour has been observed to increase as a function of RH (e.g. Ibrom et al., 2007; Mammarella et al., 2009; Nordbo et al., 2012b) and this is likely true for methanol as well. In addition, the length of the sampling tube affects the response time as well but the effect is difficult to quantify without experimental data (Nordbo et al., 2013).

The correction factor $\alpha_{\text{horst}}$ for the high frequency losses was 1.16 on average. Even though the use of the constant value of $\tau = 1.0$ s may lead to random uncertainties if the true response time varies temporally, this is likely to only have a small effect on the calculated fluxes. Also a systematic error of few percentages is possible, if the actual average response time was smaller or higher. We can also note that the change of the flow rate from 20 to 40 l min$^{-1}$ had only a negligible effect on the attenuation as long as the flow is turbulent (see Nordbo et al., 2014).

In addition to the high frequency losses and lag-time searching routines, the calculated flux values may also be biased by some other factors. For short-lived isoprene and monoterpenes (minimum lifetimes ca. 2 hours, see Hellén et al., 2012), the flux losses due to chemical degradation were estimated to be few percentages (see Rinne et al., 2012). However, these losses are difficult to compensate as they do depend on oxidant concentrations (mainly OH and $O_3$) and on surface layer mixing.

Thus, no corrections due to the chemical degradation were applied. All the flux values were slightly underestimated ($< 3\%$ based on the measured $CO_2$ fluxes) as the low frequency corrections were left out due to noisy VOC spectra. Larger errors might be produced by calibration uncertainties that affect directly the measured fluxes. All mass-to-charge ratios excluding $m/z$ 47 (ethanol+formic acid) were directly calibrated against a standard in this study. According to Kajos et al. (2015), the concentrations of the calibrated compounds may also be biased. The flux values of ethanol+formic acid should especially be

considered with caution as the concentrations of $m/z$ 47 signal was scaled based on transmission curves (see Taipale et al., 2008).

Periods when the anemometer or the PTR-MS were working improperly, were removed from the time series (Fig. 1). For example, the fluxes were not measured during summer 2013 due to a thunderstorm that broke the anemometer, and in the beginning of 2014, when the PTR-MS was serviced. During some periods, signal levels did not behave normally but had

for example a lot of spikes. Those periods were disregarded as well. To limit the underestimation of the absolute flux values caused by weak mixing, the fluxes during which $u_* < 0.2\,\mathrm{m\ s^{-1}}$ were rejected from further analysis. Other quality controls, such as filtering the flux data with the flux detection limits or with the stationarity criteria (Foken and Wichura, 1996), was not performed because applying these methods for the noisy DEC data would potentially bring other uncertainty sources. For example, disregarding the fluxes below the detection limit would lead to an overestimation of the mean absolute flux values.

However, before calculating correlation coefficients between a specific VOC and another compound (CO, $CO_2$ or another VOC), a percentage (1%) of the lowest and highest values were removed to avoid effect of possible outliers. Data coverages for VOC fluxes are listed in Table 2.

### 2.2.2 Identification of measured mass-to-charge ratios

Identifications of the measured mass-to-charge ratios are listed in Table 2. Most of the identifications are clear but there are

some exceptions. First of all, p-cymene fragments to the same $m/z$ 93 with toluene (Tani et al., 2003), therefore, p-cymene may potentially have an influence on the observed concentrations at $m/z$ 93 as the used $E/N$-ratio, 135 Td, can cause fragmentation of p-cymene (Tani et al., 2003). However, Hellén et al. (2012) observed that the p-cymene concentrations at the SMEAR III site are low compared with the toluene concentrations, being around 9% in July. Therefore, the major compound at $m/z$ 93 was likely toluene, although p-cymene might have increased the fluxes at $m/z$ 93 during warm days.

Anthropogenic furan (de Gouw and Warneke, 2007) and cycloalkanes had probably a major contribution on the measured $m/z$ 69 concentrations between October and May as isoprene concentrations at the site are reported to be small (around $5 - 30$ ppt; Hellén et al., 2006, 2012). In our study, the mean $m/z$ 69 concentrations between June and August were only ca. 60% larger than during the other seasons (Table 3), indicating a considerable influence of furan and cycloalkanes (e.g. cyclohexane, see Hellén et al., 2006 and Lee et al., 2006). Another important compound influencing the measurements at $m/z$ 69 is methylbutenol

(MBO) fragment (e.g. Karl et al., 2012). However, MBO is mostly emitted by conifers (e.g. Guenther et al., 2012) that are rare near the SMEAR III station. Therefore, MBO should only have a negligible effect on the concentration and fluxes measured at $m/z$ 69.

Monoterpenes fragment to the $m/z$ 81. The parental mass-to-charge ratio of the monoterpenes, $m/z$ 137, had a low sensitivity
during the study, and therefore, the monoterpene concentrations were calculated using $m/z$ 81. For some reason the monoterpene concentrations were only slightly higher during the summer than during the other seasons (Table 3). Therefore, a contribution of other compounds than monoterpenes at $m/z$ 81 can be possible. On the other hand, Hellén et al. (2012) observed also considerable monoterpene concentrations at the site in winter, spring and fall, possibly due to anthropogenic sources.

Acetone and propanal are both measured at $m/z$ 59 with the PTR-MS but Hellén et al. (2006) showed that the average
propanal concentrations were only around 5% compared with the average acetone concentrations in Helsinki during winter. Thus, most of the $m/z$ 59 signal consisted probably of acetone. However, as propanal fluxes at the site are unknown, $m/z$ 59 will still be referred as acetone+propanal.

Measurements at $m/z$ 107 consisted of $C_2$-benzenes including, for example, o- and p+m-xylene and ethylbenzene. According to Hellén et al. (2012), major compounds measured at the site is p+m-xylene. Other important compounds reported are o-xylene
and ethylbenzene. Hellén et al. (2012) observed annual variation for those compounds with a minimum in March. In our study, only small differences between the seasons were observed (Table 3). However, the measured concentrations in this study were quite close to the corresponding values in Hellén et al. (2012). For example, the summed concentration of o-, p+m-xylene and ethylbenzene was ca. 0.16 ppb in July (Hellén et al., 2012) whereas in this study, a mean value from June–August was 0.23 ppb (Table 3).

The mass-to-charge ratio 42 is connected with acetonitrile but Dunne et al. (2012) observed that the signal might be partly contaminated by product ions formed in reactions with $NO^+$ and $O_2^+$ that exist as trace amounts inside the PTR-MS. However, this effect was impossible to quantify in this study, and thus, $m/z$ 42 was assumed to consist of acetonitrile. Generally, acetonitrile is used as a marker for biomass burning as it is released from those processes (e.g. Holzinger et al., 1999; De Gouw et al., 2003; Patokoski et al., 2015).

**2.3  Estimating biogenic contribution of isoprene**

A well-known algorithm for isoprene emissions ($E_{\text{iso}}$) is written as

$$E_{\text{iso}} = E_{0,\text{synth}} C_T C_L, \tag{4}$$

where $E_{0,\text{synth}}$, $C_T$ and $C_L$ are the same as in the traditional isoprene algorithm (Guenther et al., 1991, 1993; Guenther, 1997). The shape of this algorithm is based on the light response curve of the electron transport activity ($C_L$) and on the temperature
dependence of the protein activity ($C_T$). The emission potential, $E_{0,\text{synth}}$, describes the emission rate of isoprene at $T = 30°C$ where $T$ is the leaf temperature (the ambient temperature in this study).

The algorithm was used to identify possible biogenic isoprene emissions. For other compounds, such as methanol or monoterpenes, no empirical algorithms were applied.

# 3 Results and discussion

## 3.1 Seasonal behaviour of observed fluxes and concentrations

Significant fluxes were observed for methanol ($m/z$ 33), acetaldehyde ($m/z$ 45), ethanol+formic acid ($m/z$ 47), acetone+propanal ($m/z$ 59), isoprene+furan+cycloalkanes ($m/z$ 69, later referred as iso.+fur.+cyc.), benzene ($m/z$ 79), toluene ($m/z$ 93), $C_2$-benzenes ($m/z$ 107) and sum of monoterpenes ($m/z$ 81). The fluxes of these compounds had also a diurnal cycle at least in one of the wind sectors (Fig. 3, Table 1). Correlation coefficients between VOC, CO, $CO_2$ fluxes and traffic rates are shown in Table A1.

Methyl *tert*-butyl ether (MTBE) and *tert*-Amyl methyl ether (TAME) are commonly connected to the vehicle exhaust emissions as the compounds were at least used to increase the octane number of gasoline (e.g. Hellén et al., 2006). MTBE and TAME were measured at their parental ions at $m/z$ 89 and $m/z$ 103, respectively. However, both mass-to-charge ratios showed no significant fluxes, and therefore, those measurements were excluded from further analysis. As the identification of these mass-to-charge ratios was uncertain, both $m/z$ 89 and $m/z$ 103 are marked as *unknown* in Table 2. Formaldehyde, which was measured at $m/z$ 31 showed no fluxes either. Therefore, $m/z$ 31 was excluded from further analysis as well.

All of the studied compounds except acetonitrile had significant fluxes during winter (Table 4), indicating anthropogenic sources. All compounds except acetonitrile, iso.+fur.+cyc. and monoterpenes had also a significant difference between weekday and weekend values (Fig. 4) which is also a strong anthropogenic signal as many anthropogenic activities can expected to be lower during the weekend than during the weekdays.

The toluene and $C_2$-benzene fluxes showed statistically significant seasonal variation with a maximum in winter and a minimum in summer–autumn (Table 4 and Fig. 5). However, the variations were rather small because the biogenic emissions of these compounds should be either small or negligible, and the anthropogenic emissions are unlikely to have large seasonal variations. Nevertheless, the traffic counts were lower during June–August (Fig. 1). The average benzene fluxes had statistically no significant differences between the seasons (Table 4 and Fig. 5). A ratio between the average toluene or $C_2$-benzene and benzene fluxes had no considerable seasonal trend either (Table A2).

The benzene and toluene concentrations had a clear annual trend with a minimum during June–August. This is a well understood pattern and it is partly caused by the different atmospheric lifetimes of these compounds between seasons (e.g. Hellén et al., 2012). Of course, local sources may affect the observed concentration trend as well if the boundary layer height has a seasonal cycle. The concentrations of all aromatic compounds had also a diurnal cycle with a maximum during morning rush hours when the traffic related emissions were high and the atmospheric boundary layer was still shallow after the night (Fig 6). The behaviour is similar compared with the CO and $CO_2$ concentrations.

A clear biogenic signal was observed for iso.+fur.+cyc. which had a large difference in both the fluxes and concentrations between winter and summer (Tables 4 – 3). Therefore, the fraction of terpenoid to the total VOC fluxes was also higher in the summer than in the winter (Fig. 7). The iso.+fur.+cyc. flux followed also well the ambient temperature (Fig. 8). The monoterpene fluxes were significantly higher during the summer but the average flux during the winter was considerable as well (Table 4), indicating other major sources than only the biogenic ones. Interestingly, both the monoterpene and iso.+fur.+cyc.

concentrations peaked during morning rush hour (Fig. 6), indicating an anthropogenic contribution, most likely from traffic related sources.

Methanol had a higher average flux during spring and summer compared with winter and autumn (Table 4 and Fig. 5). Similarly, an average acetaldehyde flux from summer was around 100% larger compared with the winter value, which might indicate a significant biogenic contribution during the summer. For methanol and acetone, the largest difference between the average fluxes was interestingly between summer and autumn season. This cannot be explained by the biogenic emissions as the autumn values were smaller than the winter ones (Table 4 and Fig. 5), but it might be a result of changes in the non-traffic related anthropogenic activity. On the other hand, the observed differences can be partly explained by wind directions: in summer, 38% of the time the wind blew from the road sector ($40° - 180°$) whereas in autumn, the corresponding occurrence was only 24%.

The methanol, acetone and acetaldehyde (OVOCs) concentrations had also a seasonal cycle with a maximum in summer. However, those compounds showed no clear diurnal cycles, probably due to high ambient background concentrations compared with aromatic or terpenoid compounds (Table 3; Fig 6). The ratio of the measured OVOC fluxes to the total measured VOC fluxes stayed stable, being 48–61% depending on the season (Fig. 7).

The diurnal concentration level of acetonitrile stayed almost constant but the concentrations showed an annual trend with a maximum in summer (Table 3 and Fig. 5). However, this was probably related to advection from distant sources (e.g. Patokoski et al., 2015). Generally, the average acetonitrile fluxes were really small being still above the detection limits except in winter (Table 4; Fig. 5).

Both ethanol+formic acid fluxes and concentrations had significant differences between the seasons. However, as the ethanol+formic acid was not calibrated, the results should be taken as rough estimates. Nevertheless, the average ethanol+formic acid flux seemed to have a maximum in winter. Their concentration showed also a weak diurnal trend with minimum during early morning (Fig. 6).

## 3.2   VOC, CO and $CO_2$ emissions from different sources

To investigate the relative contributions different sources, the fluxes were analysed by wind sectors. The data was divided into three groups based on the local wind direction corresponding to built, road and vegetation dominated areas (Table 1 and Fig. 2). The measured flux value was defined to be, for example, from the road sector if less than 30% of the flux footprint area covered other than the road sector. Thus, the periods when wind blew close to a sector border, were rejected from further analysis. The total rejection rate was around 30%. The footprints were determined according to Kormann and Meixner (2001).

The CO flux was observed to have a clear diurnal cycle, and as expected, the highest emissions were detected from the road sector (Fig. 9) where the traffic emissions are at their highest. The measured CO fluxes from the road sector also correlated very well with both the corresponding $CO_2$ fluxes ($r = 0.68$, $p < 0.001$) and with the traffic rates ($r = 0.56$, $p < 0.001$, Fig. 10). The average and median CO and $CO_2$ fluxes and CO concentrations from April 3 – May 27 2014 are presented in Table 5. The ratio between the median CO and $CO_2$ fluxes was the lowest during night-time due to respiration of $CO_2$ from vegetation (Fig. 9). The highest flux values of both CO and $CO_2$ were observed during day-time. However, the rush hour peaks cannot be

seen from the flux data. On the other hand, the traffic rates were only slightly higher during the rush hours compared with the other day-time values.

During the measurement period, the average CO flux from the road sector was ca. 0.52% compared with the corresponding $CO_2$ flux (Table 5). On the other hand, $CO_2$ probably already experienced biogenic uptake between April and May 2014 (Järvi et al., 2012; Fig. 9; Table 5). Therefore, a better estimate for the flux ratio was taken from Järvi et al. (2012) who estimated that the $CO_2$ emission rate from the road sector is 264 $\mu$g m$^{-2}$s$^{-1}$(1000 veh h$^{-1}$)$^{-1}$ which is based on wintertime data from 5 years. In our study, the corresponding CO emission rate from traffic was 0.9 $\mu$g m$^{-2}$s$^{-1}$(1000 veh h$^{-1}$)$^{-1}$ which is ca. 0.34% compared with the corresponding emission rate of $CO_2$ in mass basis. Järvi et al. (2012) used data from a more narrow wind sector, 40–120°. However, the average CO fluxes had no considerable differences between the more narrow and the whole road sector. Thus, this probably had only a minor effect on the results. The $CO/CO_2$ fraction was smaller than in previous study conducted in Edinburgh by Famulari et al. (2010) who estimated that the traffic related CO emissions are 0.60% compared with the corresponding $CO_2$ emissions in Edinburgh (in mass basis). In that study, the $CO/CO_2$ flux ratio was also otherwise quite large, 1.36%. On the other hand, Harrison et al. (2012) found that the $CO/CO_2$ flux fractions of 0.32% and 0.55% which are closer to the flux ratio obtained in this study. Furthermore, the Edinburgh data set is many years older and the traffic related anthropogenic CO emissions have generally decreased during these years (e.g. Air quality in Europe – 2015 report, http://www.eea.europa.eu//publications/air-quality-in-europe-2015, accessed 2 May 2016).

Considerable CO fluxes were observed from the built sector during afternoons (Fig. 9). Such a behaviour was not observed for the $CO_2$ during the same period (Fig. 9). Domestic burning sources might explain part of the observed behaviour of the CO fluxes from the built sector. On the other hand, many car engines are always started in the afternoon (between Monday and Friday) when people are leaving the university campus. Catalytic converters that oxidize CO to $CO_2$ may not work properly right after starting the engine (e.g. Farrauto and Heck, 1999) leading to the high observed CO emissions. Unfortunately, the CO data set from the built sector was very limited from weekends. Therefore, the CO fluxes from the working days could not be compared with the CO fluxes from Saturday and Sunday. However, aromatic VOCs seem to have a similar behaviour with increasing values during afternoon from the built sector (Fig. 3) which is somewhat expected as Reimann and Lewis (2007, p. 33) mentions that the VOC related "cold start emissions" are becoming more and more important. On the other hand, none of the aromatic compounds had a positive correlation with the CO flux, indicating different sources for CO and for the aromatic compounds.

According to a study by Hellén et al. (2006), the traffic is the most important source for the aromatic compounds in Helsinki with for example wood combusting explaining less than 1% of the detected benzene concentrations. However, the study by Hellén et al. (2006) was based on the chemical mass balance receptor model with the VOC concentrations. Thus, the footprint of their study was larger than in our work which is based on the flux measurements. The major emissions could originate also from the biogenic sources, at least in the case of isoprene and monoterpenes (Hellén et al., 2012). Therefore, summertime data of iso.+fur.+cyc., monoterpenes, and also OVOCs (methanol, acetone+propanal and acetaldehyde) were analysed more carefully. Conversely,the aromatic compounds were assumed to have no biogenic emissions, although benzenoid compounds might also originate from vegetation (Misztal et al., 2015).

In addition to the traffic, other anthropogenic VOC sources could potentially include wood combusting and solvent use. Industry is also a source for the VOCs but no industrial activities were located inside flux footprint areas. However, the solvent use might be a significant source for many compounds, especially in the built sector where the university buildings are located.

### 3.2.1 Traffic related emissions

Out of the measured compounds, methanol, acetaldehyde, ethanol, acetone, toluene, benzene, and $C_2$-benzenes are ingredients of gasoline (Watson et al., 2001; Niven, 2005; Caplain et al., 2006; Langford et al., 2009). Therefore, the traffic is potentially an important anthropogenic source for these compounds. In addition, many studies have shown traffic related isoprene emissions (Reimann et al., 2000; Borbon et al., 2001; Durana et al., 2006; Hellén et al., 2006, 2012). Hellén et al. (2012) also speculated that some of the monoterpene emissions could originate from the traffic. Of course, the ingredients of gasoline probably do have variations between countries. In Finland, a popular 95E10 gasoline contains a significant amount of ethanol ($< 10\%$) and methanol ($< 3\%$).

In recent VOC flux studies at urban sites, the fluxes of some VOCs have correlated with the traffic rates (Langford et al., 2009, 2010; Park et al., 2010; Valach et al., 2015) but this does not necessarily imply causality. At SMEAR III, the traffic has been shown to be the most important source for $CO_2$ from the road sector (Järvi et al., 2012) and the same seems to hold also for CO (Table 5). Therefore, the influence of the traffic on the VOC emissions was quantified by studying the measured VOC fluxes from this direction. The difference between the average fluxes from the road sector and the other sectors was statistically significant (95% confidence intervals) for methanol, acetaldehyde, iso.+fur.+cyc., benzene and $C_2$-benzenes.

All three studied aromatics (benzene, toluene and $C_2$-benzenes) were assumed to have same main source, the traffic. Therefore, the aromatic compounds are analysed together and they are later referred as the aromatic flux. However, especially toluene and $C_2$-benzenes are also released from solvents and paint related chemicals. These non-traffic related sources were studied by comparing the average toluene and $C_2$-benzene flux with the corresponding average benzene flux, as benzene was assumed to be emitted from the traffic related sources only. The ratios between the average toluene and benzene fluxes from the road, vegetation and built sector were around $2.6 \pm 0.4$, $2.50 \pm 0.7$ and $3.70 \pm 1.9$, respectively. The ratios indicate that toluene might have also evaporative sources. In previous studies, the exhaust emission ratio between toluene and benzene has been determined to be around $2 – 2.5$ (e.g. Karl et al., 2009 and references therein) but the ratio depends on catalytical converters etc. (e.g. Rogers et al., 2006). Above an industrialized region in Mexico City where toluene had also other major sources in addition to traffic, Karl et al. (2009) found the ratio to be around 10–15. In this study, the corresponding ratios for benzene/$C_2$-benzenes were $0.32 \pm 0.05$, $0.31 \pm 0.09$ and $0.30 \pm 0.17$. In earlier studies (e.g. Karl et al., 2009 and references therein), the exhaust emission ratio for those compounds have been observed to be around 0.4. Thus, both toluene and $C_2$-benzenes had probably also other than traffic related emissions in all the sectors. However, the possible sources for these non-traffic related emissions remained unknown. In the built sector, the evaporative emissions from the University buildings might explain part of the toluene and $C_2$-benzene flux.

The traffic rates and the aromatic fluxes had a significant correlation ($r = 0.38$, $p < 0.001$, measurements between January 2013 and September 2014) from the road sector. The aromatic fluxes correlated even better with the measured CO fluxes

($r = 0.50$, $p < 0.001$, measurements between April and May 2014). The significant correlation between the aromatic VOC flux and the CO flux indicates a common source from incomplete combustion. As these both correlated in also with the traffic rates, the traffic is likely to be the major source for aromatics.

To estimate the total emission of the aromatic compounds from the traffic, the aromatic fluxes were fitted against the traffic rates. A linear model between the traffic rates and the $CO_2$ emissions has been suggested, for example, in Järvi et al. (2012). On the other hand, Langford et al. (2010) and Helfter et al. (2011) proposed an exponential fit for the VOC and $CO_2$ emissions. Helfter et al. (2011) mention many reasons for the exponential relationship, such as an increased fuel consumption at higher traffic rates. However, Järvi et al. (2012) did not observe the exponential behaviour between the $CO_2$ fluxes and the traffic rates at the site. Therefore, a linear model was also used in this study. Additionally, the exponential relationship was tested but it brought no clear benefit compared with the linear model. The linear fit gave $F_{aro} = (28 \pm 5) \cdot 10^{-3} Tr + 10 \pm 9$ ng m$^{-2}$s$^{-1}$, where $F_{aro}$ is the flux of the aromatics (unit ng m$^{-2}$s$^{-1}$) and Tr is the traffic rate (veh h$^{-1}$). Based on this model and the traffic rates measured in 2013, the aromatic emission from traffic was estimated to be ca. $1.1 \pm 0.2$ g m$^{-2}$yr$^{-1}$ if the intercept is assumed to be indicative other than traffic-related sources. The uncertainty estimate excludes possible errors related to the calibrations and to the traffic counts. Nevertheless, the value $1.1 \pm 0.2$ g m$^{-2}$yr$^{-1}$ is around 0.01% compared with the corresponding $CO_2$ emission from the road sector (in mass basis) that was estimated using a linear model provided by Järvi et al. (2012).

The methanol fluxes were observed to correlate with the traffic rates ($r = 0.32$, $p < 0.001$, Sep–May) and with the CO fluxes ($r = 0.31$, $p = 0.001$, Apr–May 2014) in the road sector. According to a linear fit, the methanol flux values were around 20 ng m$^{-2}$s$^{-1}$ or higher when the traffic rate was close to zero (Fig. 11). This indicates that methanol had probably also other major sources than the traffic. This is also supported by the fact that the average methanol fluxes from weekend and weekdays were quite close to each other (Fig. 4), even though the traffic rates were clearly larger during the weekdays (Fig. 9). However, we were not able to identify any clear additional sources to the traffic except biogenic emissions during summer. To support our claim, Langford et al. (2010) found that the traffic counts were able to explain only a part of the observed methanol fluxes but other methanol sources remained unknown in that study as well.

The other oxygenated hydrocarbon fluxes correlated also with the traffic rates. The ethanol+formic acid fluxes were somewhat noisy and mostly close to the detection limit (Table 4) but the correlation between the measured fluxes and the traffic rates was still significant ($r = 0.19$, $p < 0.001$, Jan 2013 – Sep 2014). However, no correlation between the ethanol+formic acid and CO fluxes was found. The corresponding correlation coefficients for acetone+propanal were 0.23 ($p < 0.001$, traffic) and 0.42 ($p < 0.001$, CO). The correlation between the acetaldehyde and CO fluxes was 0.39 ($p < 0.001$) and between the acetaldehyde flux and the traffic rates 0.30 ($p < 0.001$). The methanol, acetaldehyde and acetone+propanal fluxes had also considerable correlations with each other, indicating that these compounds had probably similar sources from the road sector. The correlation coefficients between the methanol and acetaldehyde fluxes and methanol and acetone+propanal fluxes were 0.52 and 0.38, respectively ($p < 0.001$, measurements from Sep–May). The period between September and May was used instead of winter, i.e. non-growing season, to have a reasonable amount of data.

The iso.+fur.+cyc. fluxes measured during September–May had a weak but a significant correlation ($r = 0.20$, $p < 0.001$) with the traffic rates (Fig. 11). Moreover, the average iso.+fur.+cyc. flux was positive during winter (Table 4), indicating that

some of the iso.+fur.+cyc. fluxes originate from anthropogenic sources. A correlation between the iso.+fur.+cyc. fluxes and the traffic rates has also been earlier observed by Valach et al. (2015). A correlation between the iso.+fur.+cyc. and the CO fluxes was significant ($r = 0.37$, $p < 0.001$) also indicating a traffic related source. However, one should note that isoprene is also emitted from biogenic sources and this component is difficult to distinguish from the measured fluxes. If the data from winter months was only used, no relation between iso.+fur.+cyc. fluxes and the traffic rates was found. On the other hand, amount of data was also quite limited from those months (Table 4).

The monoterpene fluxes had only a weak correlation with the traffic rates ($r = 0.14$, $p = 0.001$). However, even the weak correlation might also have been a result of the increased biogenic emissions as they have a similar kind of diurnal cycle compared with the traffic rates. The biogenic influence would be possible to eliminate by dividing the monoterpene fluxes into different temperature classes, but the amount of data was too small for that kind of analysis. Thus, the possible monoterpene emissions from the traffic remained unknown, although the rush hour peak in the diurnal concentration cycle (Fig. 6) indicated traffic related emissions.

The acetonitrile fluxes had no correlation with the traffic rates. This was expected as the only considerable acetonitrile fluxes were observed from the built sector (Fig. 3). The acetonitrile emissions from the traffic should also be small compared to toluene or benzene emissions (e.g. Karl et al., 2009 and references therein).

Overall, the observed correlations were relatively low for all the VOCs. One explanation is that the fluxes were noisy, reducing therefore also the corresponding correlation coefficients. On the other hand, the low correlations may also indicate multiple sources for many of the VOCs, decreasing therefore the correlations between the fluxes and, for example, the traffic rates, and thus making the VOC source analysis very challenging.

### 3.2.2 Biogenic emissions

Nordbo et al. (2012a) observed that the urban $CO_2$ fluxes are clearly dependent on the fraction of vegetated land area in the flux footprint. Moreover, Järvi et al. (2012) observed that at our measurement site the vegetation sector is a sink for $CO_2$ during summer (see also Fig. 9). Thus, the biogenic VOC emissions could be expected to occur at the site. For iso.+fur.+cyc., the biogenic contribution was clear, and an anticorrelation ($r = -0.53$, $p < 0.001$) between the $CO_2$ and iso.+fur.+cyc. fluxes were observed from the vegetation sector during the summer. The iso.+fur.+cyc. fluxes were also affected by the ambient temperature with the small fluxes associated with the low temperatures (Fig. 8). Also the methanol fluxes had a high anticorrelation with the carbon dioxide fluxes from the vegetation sector between June and August ($r = -0.59$, $p < 0.001$), indicating a biogenic source as well.

The iso.+fur.+cyc. fluxes were fitted against the empirical isoprene algorithm (Eq. 4) to obtain the emission at standard conditions. Thus, the only free parameter in the fitting was the emission potential $E_0$. It has been shown before that the emission potential of isoprene might have a seasonal cycle with a maximum during midsummer (e.g.in the case of aspen: Fuentes et al., 1999; see also Rantala et al., 2015). However, due to a lack of data points, the fitting was done for the whole summer period (Jun–Aug) only. First, the fitting was done for each wind direction, but no considerable differences in the emission potentials between the wind directions were found. When all the data from the summer was used, the correlation between the measured

fluxes and the calculated emissions (Fig. 12) was good ($r = 0.81$), indicating that most of the measured flux at m/z 69 originated from the biogenic isoprene emissions during the summer. On the other hand, the algorithm was unable to explain some higher iso.+fur.+cyc. flux values from the road sector (Fig. 12). These values might be related to random uncertainties but they might also be, for example, a result of the traffic related emissions.

The calculated emission potential ($E_0 = 125 \pm 5$ ng m$^{-2}$s$^{-1}$) is roughly twice as high that has been measured above a pine dominated boreal forest in Hyytiälä, Southern Finland (Rantala et al., 2015), although the fraction of vegetation cover at SMEAR III is only 38–59%. However, this was expected as the urban vegetation consists of mostly broadleaved trees that are major isoprene emitters (e.g. Guenther et al., 2006). On the other hand, one should note that the emission potentials were determined above a rather heterogeneous terrain with multiple tree species (e.g. Botanical garden). Thus, a direct comparison
with the other studies should be avoided. More accurate analysis would be possible if dry leaf masses were known inside the flux footprint area. Unfortunately, this information was not available for this study. As a conclusion, the biogenic isoprene emissions explained around $80 \pm 5\%$ of the measured iso.+fur.cyc. flux in the summer (Table 6). This estimate was calculated by comparing the average iso.+fur.cyc. flux at low temperatures (Fig. 8) with the average flux in the summer (Table 4).

Methanol, acetaldehyde and acetone are also emitted from the biogenic sources (e.g. Guenther et al., 2012), and the methanol
fluxes were dependent on the ambient temperature (see supplementary material). The average methanol flux was around 30 ng m$^{-2}$s$^{-1}$ when temperature was less than 10°C indicating a biogenic contribution as the average flux was around 54 ng m$^{-2}$s$^{-1}$ in the summer (Table 4). For acetaldehyde and acetone+propanal, the corresponding average fluxes when $T < 10°C$ were around 9 and 14 ng m$^{-2}$s$^{-1}$, respectively. When comparing these values ($T < 10°C$) to the average summer time fluxes (Table 4) and taking to account the variation in data, the vegetation had a contribution of $42 \pm 8\%$, $26 \pm 8\%$ and $30 \pm 11\%$ for
the methanol, acetaldehyde and acetone fluxes during the summer, respectively. Together, the biogenic emissions explained around 35% of the total OVOC flux during summer. These estimates are valid if anthropogenic emissions are assumed to be independent of the ambient temperature. Therefore, the estimates are only rough but still reasonable. For example, the measured biogenic OVOC emissions in Hyytiälä, Southern Finland, have been comparable (Rantala et al., 2015).

The average monoterpene flux was around 7 ng m$^{-2}$s$^{-1}$ when temperature was $< 10°C$ (Fig. 8), indicating that the sig-
nificant monoterpene emissions originated from other sources than the biogenic ones. Therefore, no empirical emission algorithms were fitted against the monoterpene fluxes. Nevertheless, in June–August the average monoterpene flux was around twice as high, when compared with the average at the low temperatures ($T < 10°C$; Table 4). Taking into account the variation in the data, the biogenic contribution was assumed to be $50 \pm 15\%$ of the value of the average monoterpene emissions in the summer. Overall, the anthropogenic emissions were estimated to be around 35% compared with the total terpenoid
(isoprene+monoterpenes) emission in the summer (Table 6).

### 3.2.3   Other VOC sources or sinks

The other potential sources of VOCs, mainly wood combustion and solvent use, were found to be difficult to identify. For example, quite large acetone+propanal emissions were observed from the built sector in the afternoon (Fig. 3). These emissions might have been originating from the chemistry department near the site that uses acetone as a solvent. Recent studies (e.g.

Wohlfahrt et al., 2015 and references therein; Rantala et al., 2015; Schallhart et al., 2015) have shown that deposition might have a significant role in the OVOC exchange in some ecosystems. However, clear signals of net deposition were not observed for any of the studied OVOCs.

Nevertheless, methanol, acetaldehyde and acetone+propanal emissions were observed and they did not depedent on the ambient temperature or on the traffic rates. The methanol emissions were around 20–45 ng m$^{-2}$s$^{-1}$ from the road sector when the traffic rate was close to zero (Fig. 11). The intercept of the linear fit was larger during June–August than during September–May but the difference was statistically insignificant. When the sum of OVOCs (excluding ethanol+formic acid) was fitted together against the traffic rates (Sep–May), the intercept was $28 \pm 22$ ng m$^{-2}$s$^{-1}$ whereas the corresponding average OVOC flux was around 82 ng m$^{-2}$s$^{-1}$. If the intercept is assumed to be describe of a non-traffic anthropogenic flux, the ratio between the non-traffic related anthropogenic emissions and the total anthropogenic OVOC flux was $0.34 \pm 0.27$. The ratios between the average benzene and the average OVOC fluxes had no considerable differences between the sectors, thus the given estimate represents the whole measurements site. Hence, the other anthropogenic sources than the traffic explained $35 \pm 25\%$ of the total anthropogenic OVOC flux at the site (Table 6). This is, of course, a rough estimate, as the biogenic sources, traffic and other anthropogenic sources are difficult to distinguish from each other. Probably all of these sources have, for example, similar diurnal cycles with the minimum and maximum emissions during night and day, respectively.

Globally the aromatic compounds have other sources than traffic, such as solvent and petroleum use (Na et al., 2005; Srivastava et al., 2005; Langford et al., 2009). When considering an intercept of $10 \pm 9$ ng m$^{-2}$s$^{-1}$ of the linear fit between the aromatic fluxes and the traffic rates (Fig. 11), the emissions of the aromatic compounds from the non-traffic sources might play a role at the SMEAR III. A ratio between the intercept and the average aromatic flux from the road sector was $0.18 \pm 0.17$. Thus, the other sources than the traffic were estimated to explain $20 \pm 15\%$ of the measured aromatic fluxes (Table 6). Again, this represents the whole measurement site as the flux ratios between toluene or $C_2$-benzenes and benzene had no considerable differences between the sectors.

For the iso.+fur.+cyc. compounds, small emissions around $2 - 3$ ng m$^{-2}$s$^{-1}$ were detected (Fig. 8 and Table 4) originating from other than biogenic sources. They might be traffic-related as discussed above but they may also come from petroleum products (Langford et al., 2009). Nevertheless, the contribution of the iso.+fur.+cyc. emissions from the anthropogenic sources was relatively small during summer, with a maximum around $15 - 25\%$. The estimate was calculated by comparing the average iso.+fur.+cyc. flux at $< 10°C$ with the average iso.+fur.+cyc. flux between June and August. For monoterpenes, the anthropogenic influence was stronger but no clear sources were identified. However, monoterpenes could originate from solvents as they are for example ingredients of various cleaning products.

Acetonitrile had significant emissions only from the built sector. This indicates that the major sources of acetonitrile are not traffic related, although Holzinger et al. (2001) found weak signals for the traffic related acetonitrile emissions, and Langford et al. (2010) measured the acetonitrile fluxes that correlated with the traffic rates. On the other hand, Langford et al. (2010) mentioned that despite of the correlation, the acetonitrile sources were not known. In this study, a possible source for acetonitrile could be wood combusting in the residential area, which is located around 200–400 m from the site, and thus at the edge of the typical flux footprint area (see Ripamonti et al., 2013 and Fig. 2). On the other hand, for example Christian et al.

(2010) mentioned that the acetonitrile emissions from wood combusting are small in comparison with the other biomass burning sources. In addition, acetonitrile is released from the solvents. Thus, this might explain the observed acetonitrile flux as well. This is supported by the observed correlation between the acetone and acetonitrile fluxes from the built sector (Table A1). However, the acetonitrile fluxes were mostly noisy and close to the detection limits (Table 4), making any final conclusions
challenging.

### 3.3 Comparison of the results with previous VOC studies

Generally, the measured VOC fluxes were much lower than those reported in the previous urban VOC flux studies (Fig. 13). For example, Velasco et al. (2005) measured an order of magnitude higher methanol, acetone+propanal, toluene and $C_2$-benzene fluxes in Mexico City compared with this study. Most of the previous measurements were done in the city centres while this
study was done at the urban background site, which likely has a considerable effect on the magnitude of the VOC fluxes. For example, Reimann and Lewis (2007, p. 53) underlined the fact that the concentrations were lower in the suburban area of Zürich compared with the city centre.

For the measured $CO_2$ fluxes, intercity variations are found to be considerable (Nordbo et al., 2012a). For example, Helfter et al. (2011) measured ca. five times higher $CO_2$ fluxes in London than Järvi et al. (2012) at SMEAR III (Fig. 13) in Helsinki. The
variations in the carbon dioxide fluxes can be due to intensity of the anthropogenic activity, differences in the heating systems (central, electrical, domestic gas, coal, oil or wood fired heating systems), the types of public transport (electric buses and trams or diesel buses) etc. The relatively low VOC fluxes observed in this study are in line with the low carbon dioxide flux, both of which indicate relatively low anthropogenic intensity in the urban area. In this study, for example, the traffic related aromatic emissions were around 0.01% in comparison with the corresponding $CO_2$ emissions, and according to Valach et al. (2015), the
aromatic VOC fluxes measured in London were around 0.025% compared with the corresponding average $CO_2$ fluxes (scaled from yearly $CO_2$ budget, see Helfter et al., 2011). Hence, the VOC flux to the $CO_2$ flux ratio is in the same order of magnitude, although there is almost a one order of magnitude difference between the absolute aromatic flux values.

A fraction of urban vegetation has a strong influence on the $CO_2$ exchange (Nordbo et al., 2012a), thus a perfect correlation between the VOC and $CO_2$ fluxes cannot be expected. However, the larger $CO_2$ fluxes could indicate larger VOC fluxes as
both have common sources, such as traffic. In Figure 13 the average urban VOC fluxes reported in the literature are plotted against the corresponding average $CO_2$ fluxes. The lowest average VOC and $CO_2$ fluxes were found in Helsinki (Fig. 13). On the other hand, the largest $CO_2$ fluxes were measured in London, although the largest VOC fluxes were measured in Mexico City. The large VOC fluxes in Mexico City can be due to much older vehicle fleet, fewer catalytic converters and poorer fuel quality in Mexico City than in the UK (Langford et al., 2009). The differences in the ambient temperatures might also
affect the evaporative emissions. In Mexico City, the ambient temperature varied diurnally between 10 and 25°C (Fast et al., 2007) whereas in London, the average temperature was around 13°C during the measurements. In Mexico City, the evaporative emissions of toluene were considerable as the ratio between the average toluene and the average benzene flux was around 8 (Velasco et al., 2009). Therefore, the $CO_2$-fluxes do not of course directly correlate with the VOC fluxes as VOCs are released also from other than the burning processes.

The VOC flux composition differed between the cities (Fig. 13). Benzene was the least emitted compound in all three studies which is an expected result stemming from the development of catalytic converters and changes in fuel composition as the traffic related benzene emission have generally decreased dramatically (Reimann and Lewis, 2007, p. 33 and references therein). Otherwise, the VOC flux composition is unique for each of the measurement location.

**4   Conclusions**

We present results from the first urban VOC flux measurements in a northern city with cold winters. Out of 13 measured mass-to-charge ratios, the fluxes were observed for ten ($m/z$ 33, 42, 45, 47, 59, 69, 79, 81, 93 and 107). Previous published works have indicated the emissions of the same compounds in urban VOC flux studies. The different land use categories around the measurement site in different wind directions enabled us to analyse the different sources of various compounds differentiating

between the traffic, vegetation and residential sources.

The VOC fluxes varied as a function of season. Methanol had the highest fluxes in all seasons. The other OVOCs, toluene and $C_2$-benzenes fluxes were of the same magnitude with each other and had differences in the absolute flux values between winter and summer. The iso.+fur.+cyc. fluxes were clearly higher during the summer than during the winter, indicating a major contribution of biogenic isoprene emissions.

All compounds with the detectable fluxes illustrated contributions from anthropogenic sources at the site. The aromatic compounds originated mostly from the traffic whereas for the iso.+fur.+cyc. fluxes, the anthropogenic influence was less important. However, even the small iso.+fur.+cyc. fluxes had a relatively large influence on the iso.+fur.+cyc. concentrations during the winter when the biogenic emission is small. For monoterpenes, the anthropogenic influence was larger, being of similar magnitude with the biogenic emissions in summer. The oxygenated VOCs originated from the traffic, vegetation and

unknown anthropogenic sources, which probably included solvent use at the University campus. Generally, the magnitude of the traffic related OVOC emissions was estimated to be slightly higher compared with other anthropogenic sources. However, estimating the exact fraction was found to be difficult and uncertainties were large. Even in the urban background site, the biogenic activity had a contribution to the total annual OVOC exchange. For methanol, the biogenic emissions explained around 40% of the measured flux values during the summer.

On one hand, the measured VOC fluxes were much lower than have earlier been observed in the urban VOC flux studies. On the other hand, most of the earlier urban VOC flux studies have been carried out in dense city centres, such as in London, whereas this study was done ca. five kilometres from the Helsinki city centre in a semi-urban area. Moreover, the $CO_2$ fluxes have been observed to be relatively low at SMEAR III compared with the other urban stations. However, the variation of the $CO_2$ flux can only partly explain the variation in the VOC fluxes between the different urban areas.

The measured urban VOC fluxes showed considerable variations between the different cities both in quantity and in quality. Thus, a general parameterization for the VOC exchange in the urban areas may be challenging. However, links between the VOC emissions and $CO_2$ and the CO emission provide indication of the processes which need to be described by the parameterizations. To acquire this, a larger body of concomitant measurements of VOC, CO and $CO_2$ fluxes may be needed.

# Appendix A

**Table A1.** Correlation coefficients from each wind sector between VOC, CO, $CO_2$ fluxes and the traffic rates (Tr, only from the road sector) using all available data (one percent of the highest and the lowest values were disregarded). Insignificant ($p > 0.05$) correlation coefficients are not shown in the Table. For a comparison, the correlation coefficient between the $CO_2$ fluxes and the traffic rates was calculated from the same period with the CO fluxes (Apr–May 2014).

**Road sector**

|  | m/z 33 | m/z 42 | m/z 45 | m/z 47 | m/z 59 | m/z 69 | m/z 79 | m/z 81 | m/z 93 | m/z 107 | CO | $CO_2$ | Tr |
|---|---|---|---|---|---|---|---|---|---|---|---|---|---|
| m/z 33 | 1 | – | 0.52 | 0.31 | 0.38 | 0.33 | 0.29 | 0.21 | 0.3 | 0.33 | 0.31 | 0.31 | 0.30 |
| m/z 42 | – | 1 | – | – | – | 0.12 | – | – | – | – | – | – | – |
| m/z 45 | 0.52 | – | 1 | 0.32 | 0.44 | 0.44 | 0.31 | 0.13 | 0.33 | 0.35 | 0.39 | 0.33 | 0.30 |
| m/z 47 | 0.31 | – | 0.32 | 1 | 0.14 | 0.08 | 0.23 | 0.13 | 0.23 | 0.25 | – | 0.37 | 0.19 |
| m/z 59 | 0.38 | – | 0.44 | 0.14 | 1 | 0.31 | 0.23 | 0.19 | 0.24 | 0.34 | 0.42 | 0.15 | 0.23 |
| m/z 69 | 0.33 | 0.12 | 0.44 | 0.08 | 0.31 | 1 | 0.19 | 0.19 | 0.24 | 0.19 | 0.37 | – | 0.30 |
| m/z 79 | 0.29 | – | 0.31 | 0.23 | 0.23 | 0.19 | 1 | – | 0.26 | 0.25 | 0.35 | 0.21 | 0.17 |
| m/z 81 | 0.21 | – | 0.13 | 0.13 | 0.19 | 0.19 | – | 1 | 0.13 | 0.19 | – | 0.11 | 0.14 |
| m/z 93 | 0.3 | – | 0.33 | 0.23 | 0.24 | 0.24 | 0.26 | 0.13 | 1 | 0.38 | 0.37 | 0.37 | 0.30 |
| m/z 107 | 0.33 | – | 0.35 | 0.25 | 0.34 | 0.19 | 0.25 | 0.19 | 0.38 | 1 | 0.44 | 0.38 | 0.32 |
| CO | 0.31 | – | 0.39 | – | 0.42 | 0.37 | 0.35 | – | 0.37 | 0.44 | 1 | 0.68 | 0.56 |
| $CO_2$ | 0.31 | – | 0.33 | 0.37 | 0.15 | – | 0.21 | 0.11 | 0.37 | 0.38 | 0.68 | 1 | 0.43 |
| tr | 0.30 | – | 0.30 | 0.19 | 0.23 | 0.30 | 0.17 | 0.14 | 0.30 | 0.32 | 0.56 | 0.43 | 1 |

**Vegetation sector**

|  | m/z 33 | m/z 42 | m/z 45 | m/z 47 | m/z 59 | m/z 69 | m/z 79 | m/z 81 | m/z 93 | m/z 107 | CO | $CO_2$ |
|---|---|---|---|---|---|---|---|---|---|---|---|---|
| m/z 33 | 1 | 0.1 | 0.55 | 0.29 | 0.37 | 0.34 | 0.14 | 0.23 | 0.23 | 0.21 | 0.28 | -0.29 |
| m/z 42 | 0.1 | 1 | – | 0.08 | 0.09 | – | – | – | – | 0.09 | – | – |
| m/z 45 | 0.55 | – | 1 | 0.35 | 0.42 | 0.35 | 0.14 | 0.18 | 0.19 | 0.22 | 0.34 | -0.12 |
| m/z 47 | 0.29 | 0.08 | 0.35 | 1 | 0.25 | – | 0.19 | 0.13 | 0.19 | 0.24 | 0.18 | 0.18 |
| m/z 59 | 0.37 | 0.09 | 0.42 | 0.25 | 1 | 0.25 | 0.18 | 0.17 | 0.19 | 0.18 | 0.39 | -0.10 |
| m/z 69 | 0.34 | – | 0.35 | – | 0.25 | 1 | – | 0.13 | – | 0.18 | – | -0.44 |
| m/z 79 | 0.14 | – | 0.14 | 0.19 | 0.18 | – | 1 | 0.08 | 0.13 | 0.14 | – | – |
| m/z 81 | 0.23 | – | 0.18 | 0.13 | 0.17 | 0.13 | 0.08 | 1 | 0.13 | – | 0.20 | -0.17 |
| m/z 93 | 0.23 | – | 0.19 | 0.19 | 0.19 | – | 0.13 | 0.13 | 1 | 0.17 | 0.24 | – |
| m/z 107 | 0.21 | 0.09 | 0.22 | 0.24 | 0.18 | 0.18 | 0.14 | – | 0.17 | 1 | 0.22 | 0.09 |
| CO | 0.28 | – | 0.34 | 0.18 | 0.39 | – | – | 0.2 | 0.24 | 0.22 | 1 | 0.28 |
| $CO_2$ | -0.29 | – | -0.12 | 0.18 | -0.10 | -0.44 | – | -0.17 | – | 0.09 | 0.28 | 1 |

**Built sector**

|  | m/z 33 | m/z 42 | m/z 45 | m/z 47 | m/z 59 | m/z 69 | m/z 79 | m/z 81 | m/z 93 | m/z 107 | CO | $CO_2$ |
|---|---|---|---|---|---|---|---|---|---|---|---|---|
| m/z 33 | 1 | 0.21 | 0.45 | 0.37 | 0.30 | 0.27 | – | 0.24 | 0.32 | – | – | – |
| m/z 42 | 0.21 | 1 | 0.35 | – | 0.40 | – | – | – | 0.19 | 0.21 | – | – |
| m/z 45 | 0.45 | 0.35 | 1 | 0.48 | 0.25 | 0.19 | – | – | 0.26 | – | – | – |
| m/z 47 | 0.37 | – | 0.48 | 1 | 0.22 | – | 0.33 | – | 0.35 | – | – | – |
| m/z 59 | 0.30 | 0.40 | 0.25 | 0.22 | 1 | 0.18 | 0.27 | 0.18 | 0.32 | – | 0.75 | – |
| m/z 69 | 0.27 | – | 0.19 | – | 0.18 | 1 | 0.33 | – | 0.38 | 0.19 | – | -0.32 |
| m/z 79 | – | – | – | 0.33 | 0.27 | 0.33 | 1 | – | 0.23 | – | – | – |
| m/z 81 | 0.24 | – | – | – | 0.18 | – | – | 1 | – | – | – | – |
| m/z 93 | 0.32 | 0.19 | 0.26 | 0.35 | 0.32 | 0.38 | 0.23 | – | 1 | 0.4 | – | – |
| m/z 107 | – | 0.21 | – | – | – | 0.19 | – | – | 0.40 | 1 | – | – |
| CO | – | 0.52 | 0.60 | 0.55 | 0.53 | – | – | – | – | – | 1 | 0.49 |
| $CO_2$ | – | – | – | – | – | -0.32 | – | – | – | – | 0.49 | 1 |

**Table A2.** The average VOC fluxes from the different seasons compared with the corresponding benzene fluxes (Table 4). The values in the parenthesis represent 95% confidence intervals.

| | *m/z* 33 | *m/z* 42 | *m/z* 45 | *m/z* 47 | *m/z* 59 | *m/z* 69 | *m/z* 79 | *m/z* 81 | *m/z* 93 | *m/z* 107 |
|---|---|---|---|---|---|---|---|---|---|---|
| **Jan 2013–Sep 2014** | | | | | | | | | | |
| | 8.2 (±1.0) | 0.13 (±0.03) | 1.9 (±0.2) | 4.0 (±0.5) | 3.0 (±0.4) | 1.5 (±0.2) | 1 | 2.0 (±0.3) | 2.6 (±0.3) | 3.0 (±0.4) |
| **Winter** | | | | | | | | | | |
| | 4.8 (±1.9) | – | 0.7 (±0.3) | 6.3 (±2.3) | 2.4 (±0.9) | 0.3 (±0.2) | 1 | 1.2 (±0.6) | 2.7 (±0.9) | 3.4 (±1.2) |
| **Spring** | | | | | | | | | | |
| | 9.1 (±1.7) | 0.20 (±0.05) | 1.9 (±0.4) | 4.6 (±1.0) | 2.9 (±0.6) | 0.9 (±0.2) | 1 | 1.6 (±0.4) | 2.4 (±0.5) | 2.9 (±0.6) |
| **Summer** | | | | | | | | | | |
| | 11.2 (±2.0) | 0.13 (±0.05) | 2.4 (±0.4) | 3.0 (±0.7) | 4.3 (±0.8) | 3.0 (±0.6) | 1 | 2.9 (±0.6) | 2.7 (±0.5) | 2.9 (±0.6) |
| **Autumn** | | | | | | | | | | |
| | 6.0 (±1.8) | 0.18 (±0.08) | 2.2 (±0.7) | 3.8 (±1.2) | 2.6 (±0.9) | 1.8 (±0.6) | 1 | 2.3 (±1.0) | 2.4 (±1.1) | 2.7 (±1.2) |

*Acknowledgements.* We acknowledge the support from the Doctoral programme of atmospheric sciences, and from the Academy of Finland (ICOS-Finland 281255 and ICOS-ERIC 281250), and from the Academy of Finland through its Centre of Excellence program (Project No 272041 and 125238). We thank two anonymous reviewers for useful comments that helped improve the manuscript. We thank Alessandro Franchin, Sigfried Schoesberger, Simon Schallhart and Lauri Ahonen for their help in carrying the PTR-MS between our laboratory and the
5   measurement site. Finally, we thank all the people who made the ancillary data available.

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

**Table 1.** The table presents three sectors around the measurement site and the fraction of vegetation of each sector ($f_X$, see Järvi et al., 2014). The average $CO_2$ flux values (in carbon basis) were taken from Järvi et al. (2012).

| | $f_{paved}$ | $f_{build}$ | $f_{veg}$ | Annual $CO_2$ emissions [gC m$^{-2}$] (five-year average) |
|---|---|---|---|---|
| All | 0.36 | 0.15 | 0.49 | 1760 |
| Built (320–40°) | 0.42 | 0.20 | 0.38 | |
| Road (40–180°) | 0.39 | 0.15 | 0.46 | 3500 |
| Vegetation (180–320°) | 0.30 | 0.11 | 0.59 | 870 |

**Table 2.** The list of compounds for which the fluxes were determined for. The compound names and the formulas listed below in third and fourth column, respectively, are estimates for the measured mass-to-charge ratios (see e.g. de Gouw and Warneke, 2007). The second column shows whether a sensitivity was determined directly from the calibration or from a transmission curve (i.e. *calculated*), and which compounds were used in the calibrations. LoD shows the average limit of detection for 0.5 s measurement ($1.96\sigma$). Note that *m/z* 89 and m/z 103 were measured only during 27 June – 27 August 2014. Due to software problems, some data were lost. Those gaps are marked by superscripts $a$ and $b$ that correspond to the lost periods between 27 June–9 July 2014 and between 27 August–30 September, respectively. The second final column shows the flux data coverages for each of the compound from the whole period January 2013–September 2014.

| [*m/z*] | Calibration compound | Compound | Chemical formula | Data coverage [%] | LoD [ppt] |
|---|---|---|---|---|---|
| 31[a] | *calculated* | formaldehyde | $CH_2O$ | – | – |
| 33[a] | methanol | methanol | $CH_4O$ | 32.2 | 397 |
| 42[a] | acetonitrile | acetonitrile, alkane products | $C_2H_3N$ | 32.4 | 35 |
| 45[a] | acetaldehyde | acetaldehyde | $C_2H_4O$ | 32.6 | 141 |
| 47[a] | *calculated* | ethanol, formic acid | $C_2H_6O$, $CH_2O_2$ | 32.9 | – |
| 59[a] | acetone | acetone, propanal | $C_3H_6O$ | 37.0 | 71 |
| 69[a] | isoprene | isoprene, furan, cycloalkanes | $C_5H_8$ | 32.1 | 105 |
| 79[b] | benzene | benzene | $C_6H_6$ | 32.8 | 60 |
| 81[b] | $\alpha-$pinene | monoterpene fragments | | 28.5 | 120 |
| 89[b] | *calculated* | *unknown* | – | – | – |
| 93[b] | toluene | toluene | $C_7H_8$ | 31.7 | 295 |
| 103[b] | *calculated* | *unknown* | – | – | – |
| 107[b] | m-xylene,o-xylene | $C_2$–benzenes | $C_8H_{10}$ | 30.9 | 197 |
| 137[b] | $\alpha-$pinene | monoterpenes | $C_{10}H_{16}$ | – | – |

**Table 3.** The average and median concentrations for each of the measured VOC compound excluding *m/z* 31, *m/z* 89 and *m/z* 103. The error estimates of the average values were calculated using the equation $1.96 \cdot \sigma_{\mathrm{voc}}/\sqrt{N}$, where $\sigma_{\mathrm{voc}}$ is the standard deviation of the VOC time series and $N$ number of data points. The lower and upper quartiles are given in parenthesis after the median values, and the 95% quantile is shown as well. One percent of the lowest and the highest values were disregarded from the time series to avoid effect of possible outliers.

| | methanol | acetonitrile | acetaldehyde | ethanol+formic acid | acetone+propanal | iso.+fur.+cyc. | benzene | monoterpenes | toluene | C$_2$-benzenes |
|---|---|---|---|---|---|---|---|---|---|---|
| | VOC concentration [ppb] | | | | | | | | | |
| **Jan 2013–Sep 2014** | | | | | | | | | | |
| mean | 3.28 (±0.09) | 0.10 (±0.00) | 0.59 (±0.01) | 1.05 (±0.04) | 1.45 (±0.03) | 0.10 (±0.00) | 0.19 (±0.01) | 0.14 (±0.00) | 0.20 (±0.01) | 0.22 (±0.01) |
| median | 2.58 (1.61...4.57) | 0.09 (0.07...0.13) | 0.51 (0.36...0.76) | 0.71 (0.41...1.22) | 1.30 (0.85...1.89) | 0.08 (0.05...0.14) | 0.13 (0.08...0.25) | 0.12 (0.08...0.17) | 0.14 (0.05...0.28) | 0.18 (0.12...0.29) |
| 95% | 7.66 | 0.19 | 1.20 | 4.07 | 2.97 | 0.27 | 0.52 | 0.28 | 0.63 | 0.52 |
| $N$ | 2415 | 2431 | 2451 | 2477 | 2779 | 2412 | 2462 | 2139 | 2383 | 2319 |
| **Winter** | | | | | | | | | | |
| mean | 1.33 (±0.11) | 0.06 (±0.00) | 0.49 (±0.03) | 1.01 (±0.08) | 0.89 (±0.05) | 0.07 (±0.00) | 0.45 (±0.02) | 0.13 (±0.01) | 0.36 (±0.02) | 0.30 (±0.02) |
| median | 1.13 (0.79...1.67) | 0.06 (0.05...0.08) | 0.43 (0.34...0.60) | 0.82 (0.60...1.26) | 0.79 (0.60...1.11) | 0.06 (0.04...0.08) | 0.44 (0.34...0.57) | 0.12 (0.09...0.15) | 0.32 (0.21...0.45) | 0.26 (0.18...0.38) |
| 95% | 2.78 | 0.10 | 0.91 | 2.26 | 1.71 | 0.13 | 0.72 | 0.29 | 0.70 | 0.66 |
| $N$ | 176 | 199 | 207 | 203 | 354 | 203 | 357 | 203 | 380 | 371 |
| **Spring** | | | | | | | | | | |
| mean | 3.05 (±0.15) | 0.09 (±0.00) | 0.59 (±0.02) | 0.75 (±0.04) | 1.28 (±0.04) | 0.09 (±0.00) | 0.18 (±0.01) | 0.12 (±0.00) | 0.15 (±0.01) | 0.18 (±0.01) |
| median | 2.18 (1.46...3.81) | 0.08 (0.06...0.11) | 0.53 (0.40...0.73) | 0.65 (0.35...1.00) | 1.08 (0.83...1.58) | 0.08 (0.05...0.11) | 0.15 (0.10...0.23) | 0.11 (0.08...0.15) | 0.12 (0.06...0.19) | 0.15 (0.11...0.22) |
| 95% | 8.20 | 0.15 | 1.10 | 1.92 | 2.61 | 0.19 | 0.41 | 0.22 | 0.45 | 0.41 |
| $N$ | 874 | 876 | 905 | 891 | 915 | 879 | 892 | 853 | 892 | 859 |
| **Summer** | | | | | | | | | | |
| mean | 4.27 (±0.17) | 0.12 (±0.00) | 0.60 (±0.02) | 1.15 (±0.07) | 1.88 (±0.05) | 0.14 (±0.01) | 0.11 (±0.00) | 0.15 (±0.01) | 0.12 (±0.01) | 0.23 (±0.01) |
| median | 3.88 (2.40...5.70) | 0.12 (0.09...0.15) | 0.50 (0.35...0.79) | 0.79 (0.52...1.40) | 1.70 (1.24...2.42) | 0.12 (0.07...0.18) | 0.09 (0.06...0.14) | 0.13 (0.09...0.18) | 0.08 (0.02...0.18) | 0.19 (0.13...0.29) |
| 95% | 8.56 | 0.21 | 1.22 | 4.00 | 3.54 | 0.32 | 0.26 | 0.29 | 0.44 | 0.50 |
| $N$ | 748 | 751 | 756 | 778 | 863 | 743 | 938 | 823 | 826 | 823 |
| **Autumn** | | | | | | | | | | |
| mean | 2.95 (±0.13) | 0.11 (±0.00) | 0.62 (±0.03) | 1.36 (±0.13) | 1.41 (±0.05) | 0.10 (±0.01) | 0.13 (±0.01) | 0.16 (±0.01) | 0.35 (±0.04) | 0.25 (±0.02) |
| median | 2.58 (1.62...3.96) | 0.11 (0.06...0.15) | 0.50 (0.29...0.92) | 0.60 (0.26...1.65) | 1.46 (0.67...1.91) | 0.06 (0.04...0.14) | 0.10 (0.07...0.16) | 0.14 (0.09...0.21) | 0.30 (0.08...0.5) | 0.21 (0.10...0.35) |
| 95% | 5.81 | 0.19 | 1.29 | 4.36 | 2.64 | 0.28 | 0.31 | 0.36 | 1.03 | 0.62 |
| $N$ | 617 | 605 | 583 | 605 | 647 | 587 | 275 | 260 | 285 | 266 |

**Table 4.** The average and median fluxes for each measured VOC compound excluding $m/z$ 31, $m/z$ 89 and $m/z$ 103. The error estimates of the average values were calculated using the equation $1.96 \cdot \sigma_{\mathrm{voc}}/\sqrt{N}$, where $\sigma_{\mathrm{voc}}$ is the standard deviation of the VOC time series and $N$ number of data points. The lower and upper quartiles are given in parenthesis after the median values. One percent of the lowest and the highest values were disregarded from the time series to avoid effect of possible outliers. The mean detection limits ($\overline{\mathrm{LoD}}$) were calculated as $\overline{\mathrm{LoD}} = 1/N \sum \mathrm{LoD}_i^2$ (Valach et al., 2015) where single detection limits, LoD, were defined to be $1.96\sigma_{\mathrm{ccf}}$ where $\sigma_{\mathrm{ccf}}$ is the standard deviation of cross covariance tails (Taipale et al., 2010). The acetonitrile flux was below $\overline{\mathrm{LoD}}$ in the winter.

| | methanol | acetonitrile | acetaldehyde | ethanol+formic acid | acetone+propanal | iso.+fur.+cyc. | benzene | monoterpenes | toluene | C₂-benzenes |
|---|---|---|---|---|---|---|---|---|---|---|
| | VOC flux [ng m$^{-2}$s$^{-1}$] | | | | | | | | | |
| **Jan 2013–Sep 2014** | | | | | | | | | | |
| mean | 44.9 (±2.5) | 0.7 (±0.1) | 10.1 (±0.6) | 21.9 (±1.7) | 16.7 (±1.1) | 8.0 (±0.6) | 5.5 (±0.6) | 10.9 (±1.2) | 14.1 (±1.1) | 16.4 (±1.4) |
| median | 29.4 (10.4...62.2) | 0.7 (-1.2...2.2) | 8.3 (2.4...16.7) | 16.5 (-0.8...35.2) | 11.6 (2.1...25.9) | 5.5 (-0.7...14) | 4.6 (-2.2...11.2) | 11.0 (-5.7...25.6) | 11.4 (-1.3...26) | 14.6 (-3.7...33.0) |
| $\overline{\mathrm{LoD}}$ | 1.2 | 0.1 | 0.5 | 1.0 | 0.7 | 0.3 | 0.5 | 0.9 | 0.7 | 1.0 |
| $N$ | 2021 | 2034 | 2050 | 2066 | 2311 | 2018 | 2090 | 1820 | 2029 | 1983 |
| **Winter** | | | | | | | | | | |
| mean | 35.5 (±7.9) | – | 5.0 (±1.3) | 46.4 (±8.8) | 17.4 (±3) | 2.4 (±1.4) | 7.3 (±2.3) | 8.5 (±3.4) | 19.6 (±3.1) | 24.6 (±4.0) |
| median | 16.4 (4.8...42.4) | – | 5.1 (-0.6...9.7) | 26.7 (8.2...69.9) | 11.6 (3.0...27) | 2.4 (-1.9...5.7) | 5.9 (-6.3...20.6) | 8.1 (-4.5...19.7) | 15.5 (1.3...35.2) | 23.3 (0.9...43.8) |
| $\overline{\mathrm{LoD}}$ | 3.0 | 0.3 | 1.3 | 4.0 | 2.5 | 1.0 | 2.2 | 3.5 | 2.5 | 3.5 |
| $N$ | 178 | – | 185 | 179 | 327 | 182 | 315 | 181 | 328 | 324 |
| **Spring** | | | | | | | | | | |
| mean | 52.1 (±4.7) | 0.9 (±0.2) | 10.7 (±1.1) | 26.2 (±3.3) | 16.4 (±1.8) | 4.9 (±0.8) | 5.7 (±1.0) | 8.9 (±2.0) | 13.9 (±1.9) | 16.5 (±2.4) |
| median | 31.8 (10.6...75.5) | 0.8 (-1.3...2.5) | 8.3 (1.7...18.6) | 18.4 (-2.2...43) | 11.2 (1.1...25.9) | 3.8 (-2.6...11.3) | 5.0 (-2.7...12.2) | 8.4 (-11.5...26.4) | 11.7 (-4.0...27.3) | 15.2 (-5.4...33.7) |
| $\overline{\mathrm{LoD}}$ | 2.3 | 0.2 | 1.1 | 2.0 | 1.3 | 0.6 | 0.8 | 1.6 | 1.1 | 1.7 |
| $N$ | 758 | 765 | 775 | 765 | 789 | 762 | 775 | 731 | 778 | 755 |
| **Summer** | | | | | | | | | | |
| mean | 54.2 (±4.5) | 0.6 (±0.2) | 11.8 (±1.0) | 14.7 (±2.2) | 20.6 (±2.1) | 14.3 (±1.4) | 4.8 (±0.8) | 14.1 (±1.8) | 13.2 (±1.5) | 14.0 (±1.9) |
| median | 39.1 (15.7...76.0) | 0.8 (-1.1...2.0) | 9.4 (3.9...18.1) | 14.1 (-2.4...29.3) | 15.0 (4.4...30.2) | 9.1 (2.1...22.6) | 4.0 (-1.1...9.3) | 12.8 (0.4...26.1) | 10.8 (2.6...22.0) | 13.1 (-2.4...29.4) |
| $\overline{\mathrm{LoD}}$ | 2.3 | 0.1 | 0.6 | 1.2 | 1.2 | 0.7 | 0.5 | 1.4 | 0.9 | 1.5 |
| $N$ | 623 | 622 | 626 | 643 | 710 | 608 | 782 | 689 | 688 | 688 |
| **Autumn** | | | | | | | | | | |
| mean | 24.4 (±2.5) | 0.7 (±0.3) | 9.0 (±1.0) | 15.6 (±2.5) | 10.8 (±2.1) | 7.3 (±1.0) | 4.1 (±1.2) | 9.3 (±3.0) | 10.0 (±3.6) | 11.2 (±3.5) |
| median | 21.4 (6.3...42.6) | 0.7 (-1.1...2.2) | 8.4 (2.7...14.3) | 14.6 (-1.5...31.2) | 7.4 (-1.2...17.9) | 5.8 (-0.1...12.6) | 4.1 (-1.4...8.5) | 9.6 (-6.0...25.8) | 7.0 (-6.6...22.0) | 8.5 (-7.3...26.9) |
| $\overline{\mathrm{LoD}}$ | 1.8 | 0.1 | 0.4 | 1.2 | 0.9 | 0.4 | 0.7 | 1.8 | 3.0 | 2.0 |
| $N$ | 462 | 465 | 464 | 479 | 485 | 466 | 218 | 219 | 235 | 216 |

**Table 5.** The statistics of the measured CO and $CO_2$ fluxes and the CO concentrations from each wind sector (3 Apr – 27 May 2014). The error estimates of the average values were calculated using the equation $1.96 \cdot \sigma / \sqrt{N}$, where $\sigma$ is the standard deviation of the CO or $CO_2$ time series and $N$ the number of data points. The lower and upper quartiles are given in parenthesis after the median values.

| | All | Built | Road | Vegetation |
|---|---|---|---|---|
| **CO flux [$\mu$g m$^{-2}$s$^{-1}$]** | | | | |
| mean | 0.69±0.05 | 0.57±0.11 | 1.46±0.15 | 0.35±0.03 |
| median | 0.36 (0.11 – 0.86) | 0.37 (0.22–0.75) | 1.18 (0.54 – 2.08) | 0.26 (0.10 – 0.48) |
| **$CO_2$ flux [$\mu$g m$^{-2}$s$^{-1}$]** | | | | |
| mean | 138±9 | 157±34 | 282±27 | 71±9 |
| median | 111 (57 – 198) | 123 (68–177) | 257 (135 – 378) | 80 (31 – 123) |
| **CO concentration [ppb]** | | | | |
| mean | 146.5±1.0 | 152.7±5.6 | 152.6±1.9 | 143.1±1.1 |
| median | 142.0 (133.8 – 155.9) | 141.2 (132.8–164.4) | 148.2 (138.7–161.4) | 139.2 (131.8 – 151.9) |

**Table 6.** The estimated contributions (%) of the aromatic and biogenic sources for the OVOCs (methanol+acetaldehyde+acetone), aromatics (benzene+toluene+$C_2$-benzenes) and terpenoids (iso.+fur.+cyc.+monoterpenes). One should note that furan and cycloalkanes may affect also to the contributions of the terpenoids. For the terpenoids, separating the different anthropogenic sources was not possible. In the case of OVOCs and aromatics, the ratio between the traffic related and the other anthropogenic emissions was assumed to have constant annual cycle.

| | OVOCs [%] | aromatics [%] | terpenoids [%] |
|---|---|---|---|
| **Winter** | | | |
| Traffic | 65 ± 25 | 80 ± 15 | – |
| Other anthropogenic sources | 35 ± 25 | 20 ± 15 | – |
| **Total anthropogenic** | 100 | 100 | 100 |
| **Total biogenic** | 0 | 0 | 0 |
| **Summer** | | | |
| Traffic | 42 ± 16 | 80 ± 15 | – |
| Other anthropogenic sources | 23 ± 16 | 20 ± 15 | – |
| **Total anthropogenic** | 65 ± 6 | 100 | 35 ± 8 |
| **Total biogenic** | 35 ± 6 | 0 | 65 ± 8 |

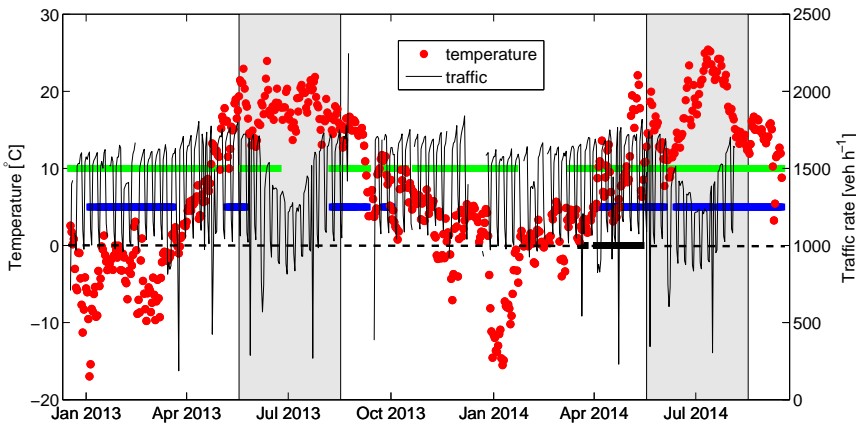

**Figure 1.** The daily averages of the ambient temperatures and the traffic rates. The data coverages of the PTR-MS (VOCs), Li-Cor 7000 (CO$_2$) and LGR (CO) measurements are marked by blue, green and black lines, respectively. The grey shaded areas show periods between June–August. The black dashed line represents the zero line of the ambient temperature.

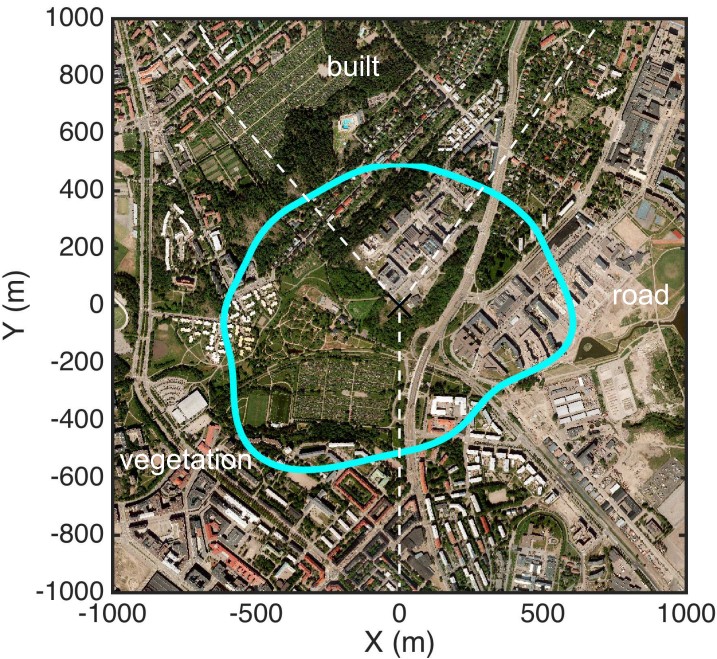

**Figure 2.** The aerial photograph of the SMEAR III station (©Kaupunkimittausosasto, Helsinki, 2011). The measurement tower is marked with a black cross. The white dashed lines represent different sectors (built, vegetation, road). The turquoise solid line shows borders of cumulative 80% flux footprint (Kormann and Meixner, 2001).

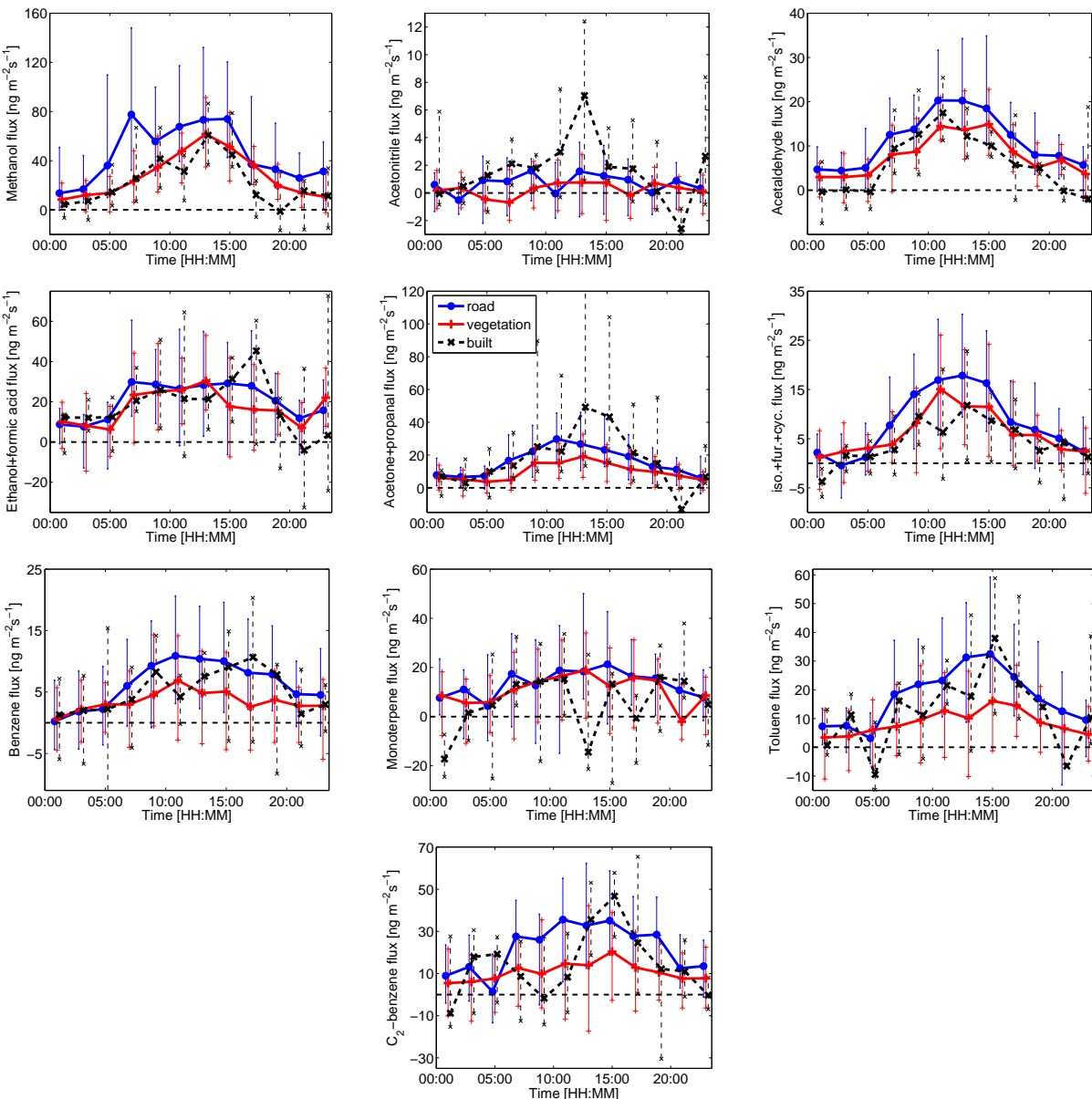

**Figure 3.** The median diurnal VOC fluxes from the three sectors for each of the compound (Jan 2013 – Sep 2014). The blue circles, red crosses and black crosses correspond to the road sector, the vegetation sector and the built sector, respectively. The vertical lines show the lower and upper quartiles (25% and 75%). Due to scaling, one upper quartile value is not shown in the acetone+propanal figure.

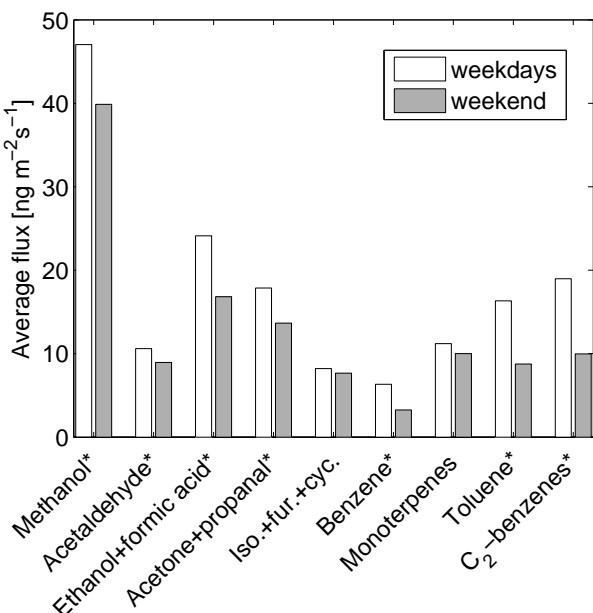

**Figure 4.** The average fluxes for each of the VOCs (excluding acetonitrile) from Saturday+Sunday and from weekdays (Jan 2013 – Sep 2014). The white and the grey bars show the average fluxes during the weekdays and Saturday+Sunday, respectively. The asterisks in the x-axes show if the differences between the average week and the average weekend fluxes were statistically significant. The uncertainties of the average fluxes were calculated using the equation $\pm 1.96\sigma_{\mathrm{voc}}/\sqrt{N}$, where $\sigma_{\mathrm{voc}}$ is the standard deviation of a VOC flux time series and $N$ the number of data points.

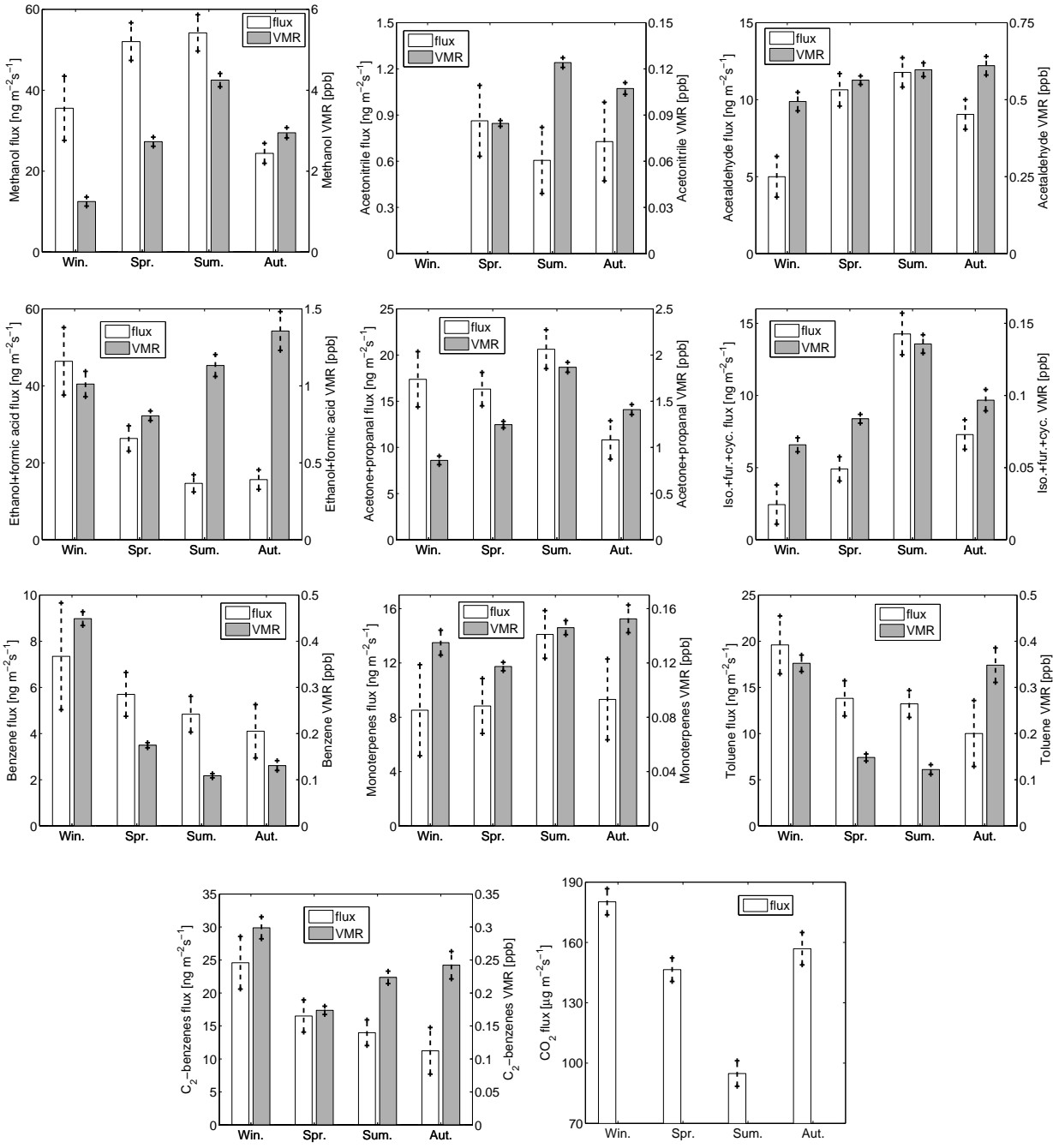

**Figure 5.** The mean seasonal flux and concentration (VMR) values for the VOCs (Tables 4 and 3). The vertical lines show the 95% confidence intervals. The seasonal cycle of the $CO_2$ flux is shown for a comparison. However, the longer gaps without the PTR-MS measurements (Fig. 1), were removed also from the corresponding $CO_2$ data.

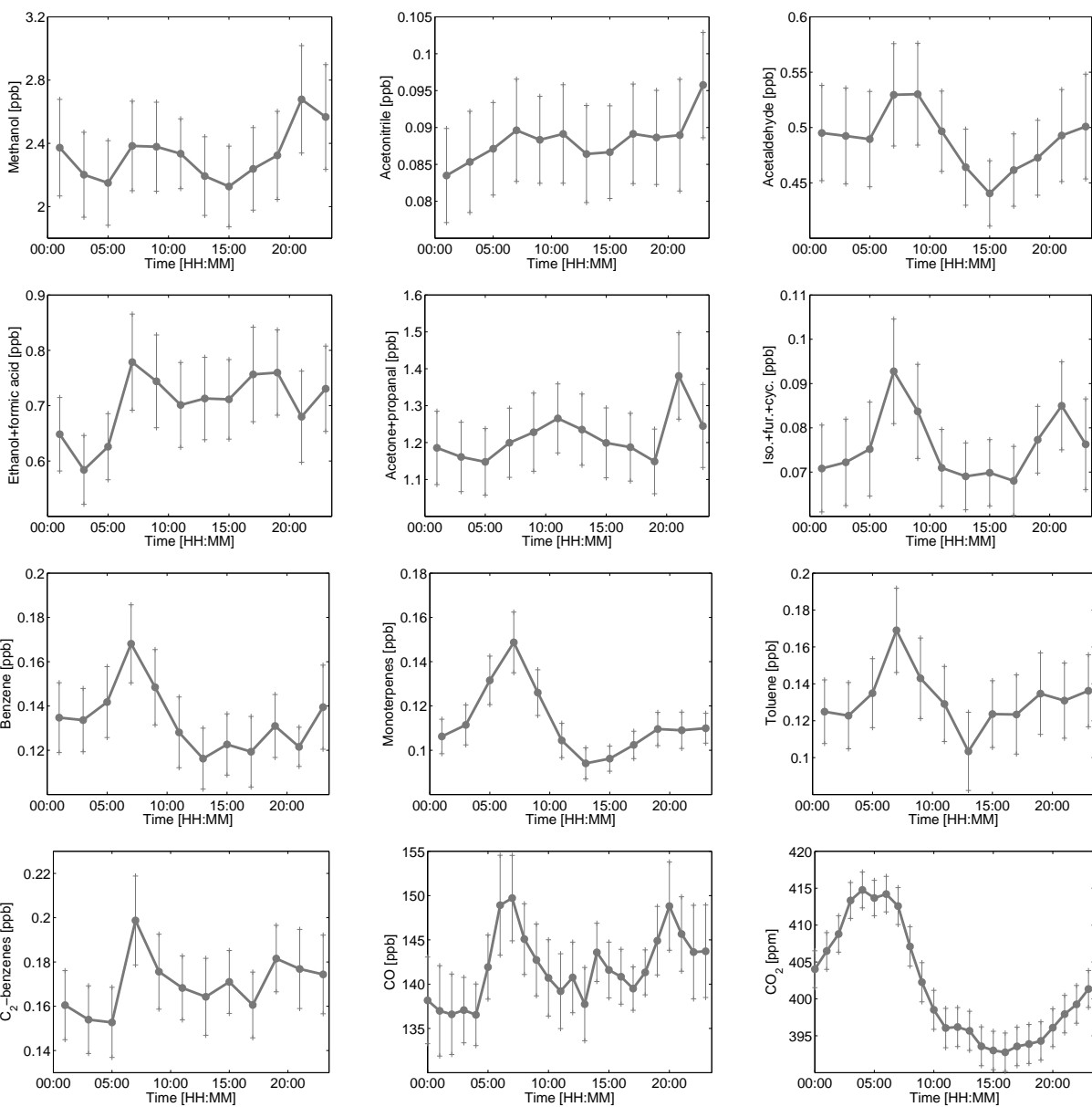

**Figure 6.** The median diurnal VOC, CO and $CO_2$ volume mixing ratios for each compound. The vertical lines show the 95% confidence intervals. The VOC and $CO_2$ data is between January 2013 and September 2014. However, times corresponding to the longer gaps in the PTR-MS data (Fig. 1), were removed also from the $CO_2$ data. The CO data is from April – May 2014.

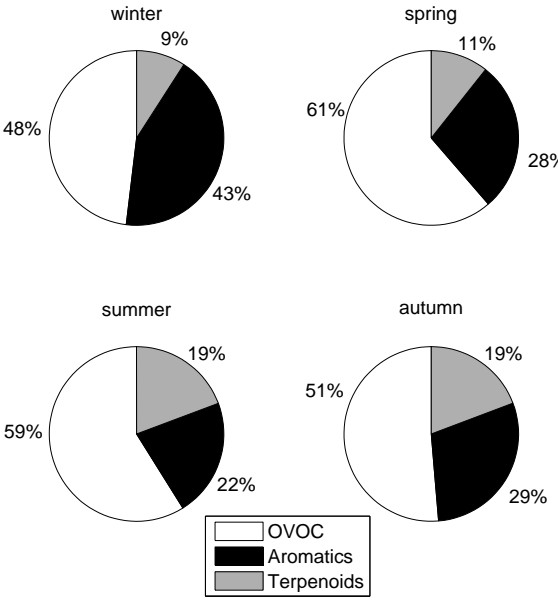

**Figure 7.** The fractions of the measured OVOC (methanol, acetaldehyde, acetone+propanal), aromatic (benzene, toluene, $C_2$-benzenes) and terpenoid (isoprene+furan+cycloalkanes, monoterpenes) fluxes from each season (in mass basis). Ethanol+formic acid was left out from the analysis as its concentrations were not directly calibrated.

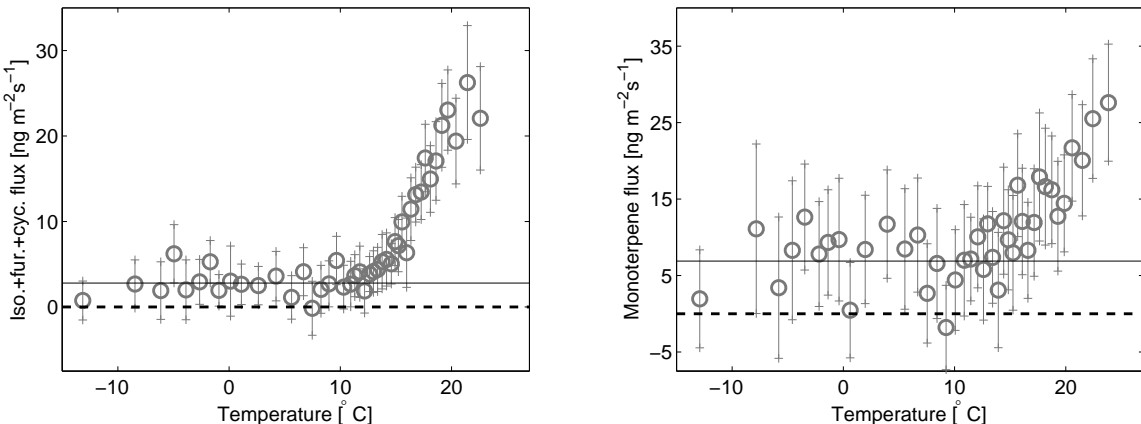

**Figure 8.** The bin-averaged iso.+fur+cyc. ($n = 45$) and monoterpene ($n = 45$) fluxes as a function of the ambient temperature (January 2013 – Sep 2014). The solid and dashed lines show the average fluxes in the range of $T < 10°C$ and zero lines, respectively.

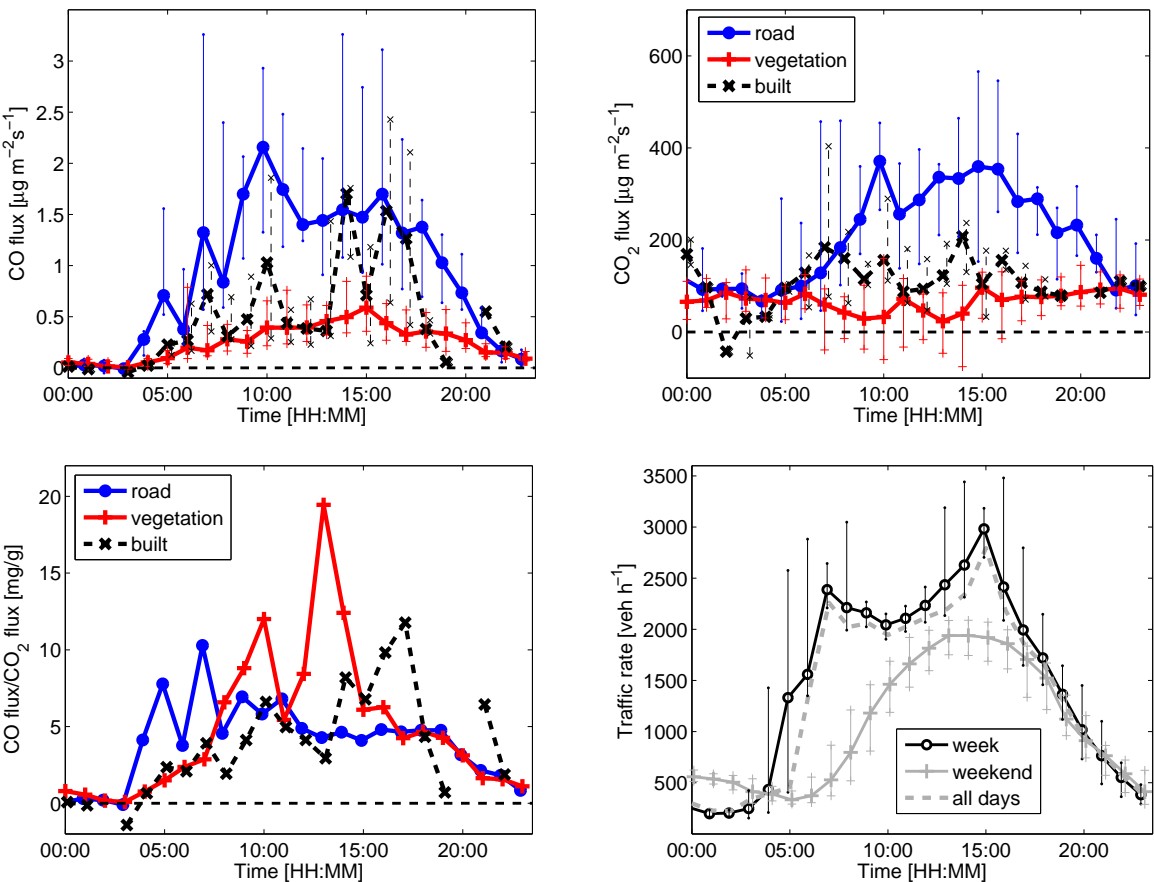

**Figure 9.** The two topmost figures present the hourly median diurnal fluxes of CO and $CO_2$ from the three sectors (3 Apr – 27 May 2014). The blue circles, red crosses and black crosses correspond to the road sector, the vegetation sector and the built sector, respectively. The vertical lines show the 25 and 75 quartiles. The ratios between the median CO and $CO_2$ fluxes are shown in the figure in the left corner. The figure in the right corner depicts the median diurnal cycles of the traffic rates from Saturday+Sunday, weekdays, and all days (Jan 2013 – Sep 2014). The vertical lines show the lower and upper quartiles for the weekend and week day values. The $CO_2$ flux is positive during night-time due to biogenic respiration (Järvi et al., 2012).

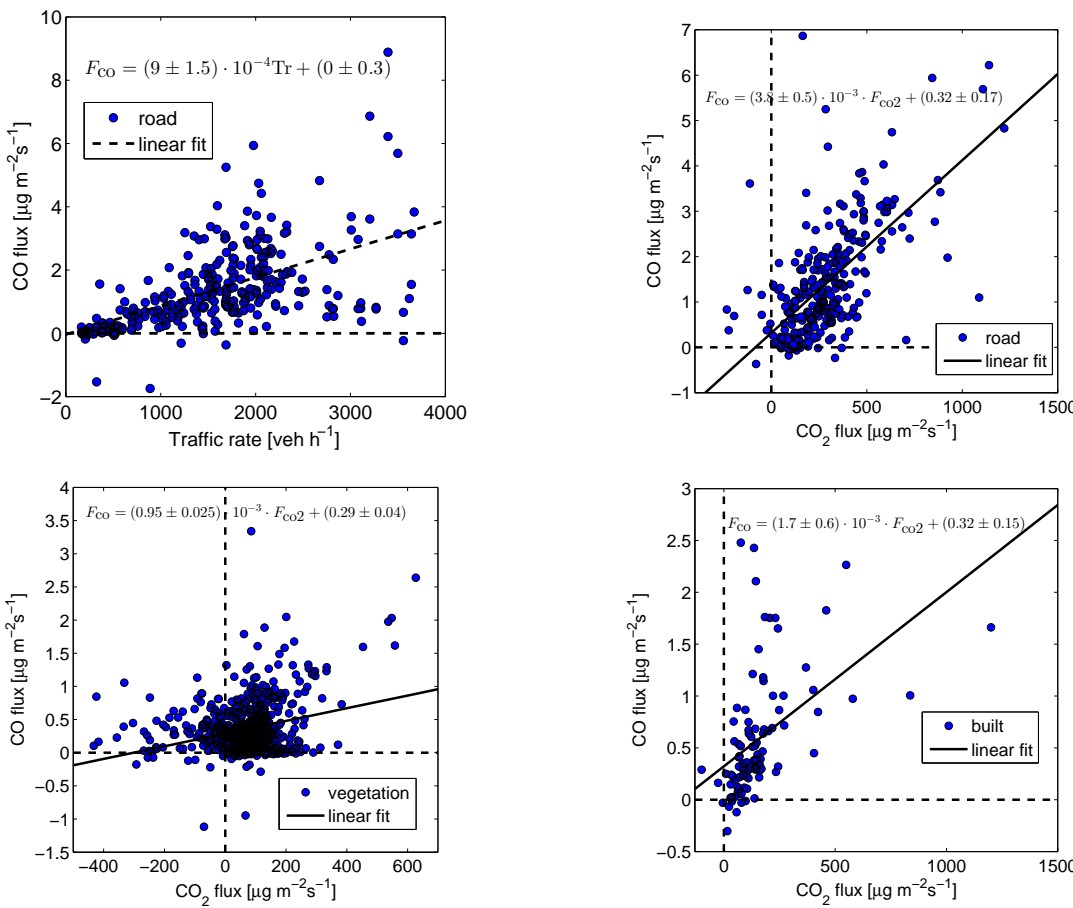

**Figure 10.** The CO fluxes against the traffic rates and the $CO_2$ fluxes from the road, vegetation and built sector (measured during April–May 2014).

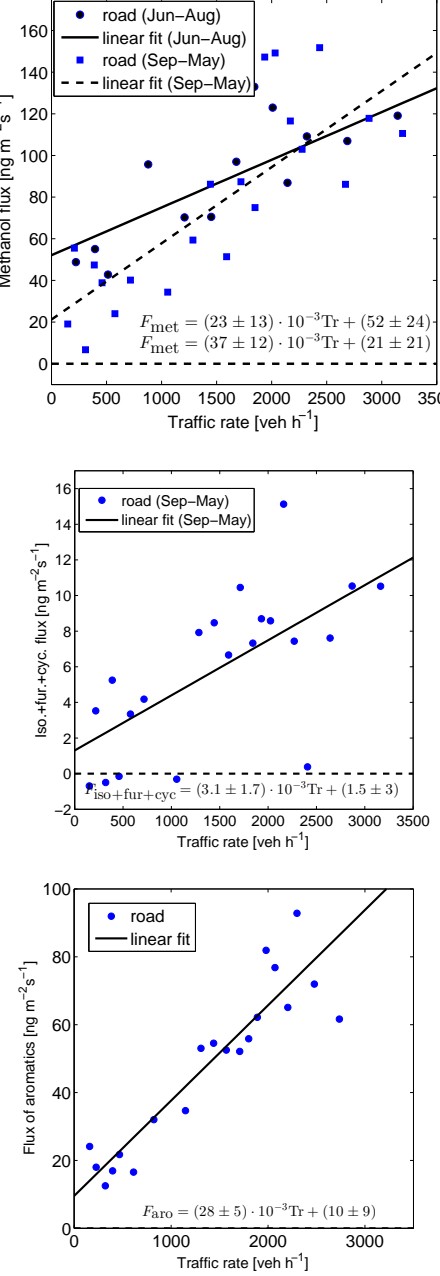

**Figure 11.** The traffic rates against the methanol (bin-averages, $n = 15$), iso.+fur.+cyc. (bin-averages, $n = 15$) and aromatic fluxes (benzene+toluene+$C_2$-benzenes, bin-averages, $n = 30$) from the road section. The linear correlations between the methanol, iso.+fur.+cyc. and aromatic fluxes and the traffic rates were 0.24 (Jun–Aug)/0.32 (Sep–May), 0.20 and 0.38, respectively ($p < 0.001$).

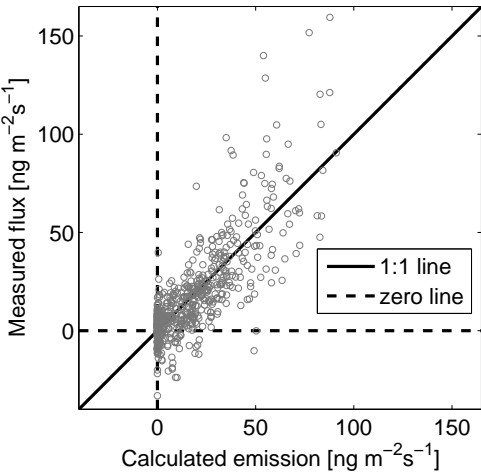

**Figure 12.** The measured iso.+fur.+cyc. fluxes vs. the calculated isoprene emissions (Eq. 4) from summer (Jun–Aug) data.

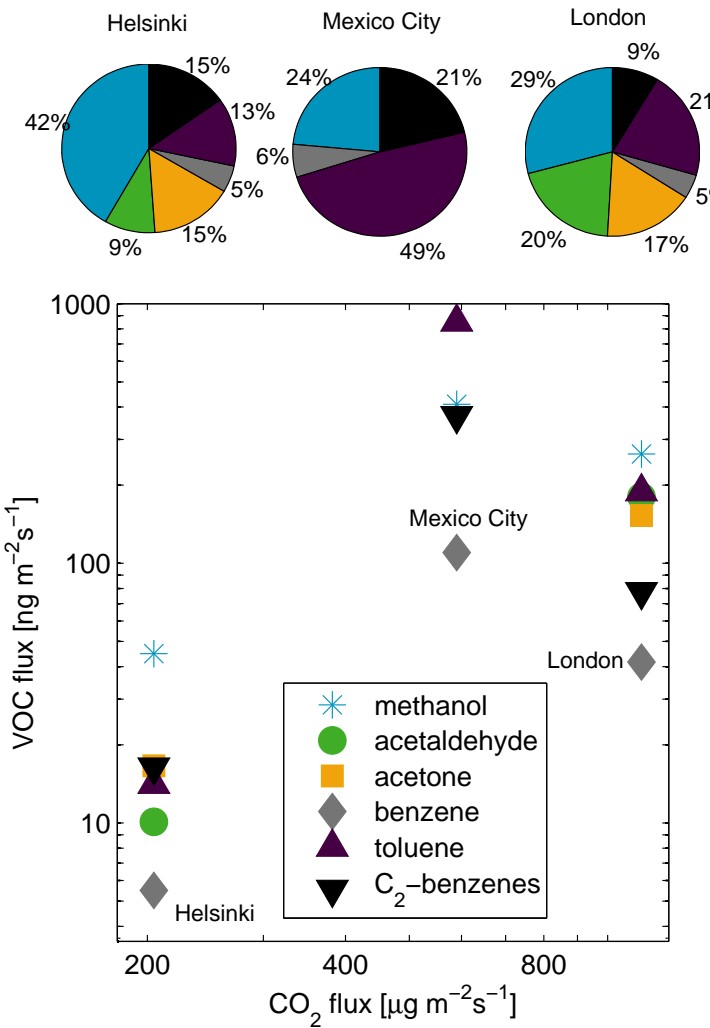

**Figure 13.** The selected VOC fluxes as a function of $CO_2$ fluxes from Helsinki, London and Mexico City (note the logarithmic scale). The average $CO_2$ and VOC fluxes for Helsinki are taken from Järvi et al. (2012) (scaled from the annual average) and this study, respectively. The corresponding average values for London are from Helfter et al. (2011) (scaled from the annual average) and Langford et al. (2010). All the values for Mexico City are from the MILAGRO/MCMA-2006 campaign (Velasco et al., 2009). The pie diagrams show the corresponding fractions of each compound.