# Peer review of "Anthropogenic and biogenic influence on VOC fluxes at an urban background site in Helsinki, Finland"

_Atmospheric Chemistry and Physics, 2015_

## Referee Comment (RC1) · Anonymous Referee #2 · 23 Mar 2016

General Comments:

This paper describes >1.5 year long-term flux measurements of VOC, CO and CO2 in the urban environment of Helsinki, Finland. As was shown in the paper and also expected for an urban environment, most VOCs have large traffic related emissions, but other anthropogenic sources are also important for some VOCs. In addition, during summer biogenic isoprene and monoterpene emissions and CO2 uptake are evident in the data. Overall VOC fluxes in the specific location of these measurements were rather small compared to other cities. Long-term flux measurements, especially in an urban area, have not been reported in the literature very often and therefore this dataset is very interesting and unique and I think a dataset like this is worth exploring

and publishing, but the analysis presented here needs major improvements before it is acceptable.

Major Issues:

1 The organization of the discussion section: I found this paper very hard to read, because of a constant mix of topics in the first part of the discussion. I would suggest to re-organize the chapters 3.1 and 3.2., before discussing the individual emission sources (traffic, biogenic, and others). The seasonal and diurnal cycles for all VOCs, CO and CO2 should be discussed in detail first then discuss individual sources. I would like to see an actual figure showing the annual cycle for VOC, CO and CO2 fluxes, although there might not be enough data for CO. Right now this important information is hidden in various figures and tables. For this discussion the data should not be separated into the three sectors. After describing these general trends in the fluxes, each emission source sector can be described: traffic, biogenic, and others; and for all of these CO and CO2 should be included and not be shown in a separate chapter.

2 It would also be important to add the mixing ratios to the annual and diurnal cycles. From the paper as is, it is not possible to understand, if this is a heavily polluted location or not. I would assume that in the cold winter months, when the boundary layer is really low, mixing ratios could get rather high.

3 The separation of the data into the three wind direction sectors looks like a good approach, when looking at the map and the potential emissions from those sectors, but the VOC flux data (Figure 3, 4 and 5) are actually very similar for each sector. The only substantial difference was found for CO and CO2 in Figure 6 and for the weekday/weekend plot in Figure 9, although that is mainly due to the traffic counts between weekday and weekend. The wind sector separation complicates the discussion in many places, but doesn't really add any information, so I think Figures 3, 4, 5, 9 and 10 should be simplified by using all the data. In addition, throughout the manuscript it becomes clear that even in the road sector substantial non-traffic related emissions are

evident and in all three wind direction sectors multiple sources contribute to the VOC emissions. This makes a quantitative analysis and separation of sources very difficult and this should be acknowledged clearly in the manuscript.

4 For some VOCs an attempt for a quasi source apportionment was done in the paper. For example, on page 13 line 10-15, the monoterpene sources are summarized and biogenic contribution was assumed to be around 40%. This type of information is to me one of the most relevant results of this paper. Unfortunately this estimation of a source apportionment was only done for monoterpenes and OVOCs and it would be important to do this carefully for all the measured VOCs. If this is possible with the data, I would like to see something like a pie chart for each VOC or class of VOCs showing the traffic, biogenic and other anthropogenic contributions for summer and winter, which should then be presented as the main result of this paper.

Specific comments:

- page 2 line 26: It would be really helpful to add the typical footprint to Figure 2. The discussion about the wind sector analysis would be much easier to follow.

- page 3 line 7: I assume this is 0.5s per mass per measurement cycle of about 6s each?

- page 3 line 8-9: Were there any other masses with significant signal or was most of the VOC signal captured by the masses used for the presented flux measurements. Please indicate other important masses.

- page 3 line 19-21: The PTR-MS instrument settings are described here, but the actual detection limits for the 0.5s measurements and the 20-30 min flux calculations are not given. Please add those, especially taking the issues with the instrument background measurements into account.

- page 3 line 24: What do you mean by "correct primary ion signal"? Is this mass discrimination corrected? Shouldn't the calibration be done at the same settings as the

actual measurements and not with optimized SEM voltages?

- page 3 line 25: Does the zero air generator change the humidity? Background measurements at a different humidity can produce significant artifacts.

- page 3 line 31-32: How much does the uncertain zero air measurement add to the uncertainty, please be specific.

- page 4 line 9: Also for CO and $CO_2$ measurements it would be good to add the precision and uncertainties.

- page 4 line 16 and page 5 line 11: Why did you use different averaging times for CO and $CO_2$ with 30 min compared to the VOCs with 45 min?

- page 5 line 13: Was there a reason not to use something like a time server synchronization program?

- page 5 line 14: Was m37 the highest flux, higher than methanol? Is that why m37 was used for the time lag calculation?

- page 6 line 31: Are those data coverages for flux measurements or do those include the mixing ratio measurements?

- page 7 lines 8-14: There have been a few recent papers about oil and gas emissions using PTR-MS showing that m69 can also have a significant influence from cycloalkanes.

- page 7 line 19: Are those anthropogenic monoterpene likely from the sector "solvents and other products" or more traffic related?

- page 7 line 33: Mention here that acetonitrile is often used as a tracer for biomass burning.

- page 8 line 6: Often biogenic inventories do not represent urban environments well, please explain where you get the E0,synth values for the measurement location from

and what your confidence in this value is.

- page 8 line 13: In Figure 3 it can be seen that acetonitrile and acetone seem to be emitted from the built sector. Could those be the result of solvent use at the University buildings?

- page 8 line 26: The annual trend in the concentration of the aromatics will also strongly depend on the boundary layer height. Atmospheric background mixing ratios of benzene are much higher in winter in the northern hemisphere with over 100ppt, but local enhancements in an urban area are probably more driven by the boundary layer height than lifetimes. Again it would be very helpful to look at annual and diurnal cycles of mixing ratios in detail as well.

- page 9 line 19-23: I think it is problematic to compare CO/CO2 ratios with other studies without taking the strong decreasing trend of CO into account. Over the past decade(s) CO and VOCs have decreased by several percent every year. The discussion should take this trend into account. Also, there are other sources of CO in a city compared to cars driving in a tunnel, e.g cold starts, (as mentioned in the text), domestic burning and other residential and commercial combustion sources. I would therefore delete the comparison with the tunnel study and look at other papers that show CO/CO2 enhancement ratios.

- page 9 lines 24-31: I agree that cold starts are likely an important source of CO and VOCs in the built sector, but I am wondering if the high CO and aromatics emissions in the afternoon could also be explained by domestic burning. Acetonitrile is generally used as a tracer for biomass burning, but it is not a good tracer for domestic burning, because N emissions are generally smaller from wood than foliage burning (e.g. Yokelson et al ACP 2014). So the lack of acetonitrile fluxes by itself is not a reason to discard domestic burning as a major source of CO and VOCs in winter. The domestic burning should have a strong annual cycle. I am not sure, if without the annual cycle measurement of CO, there is enough evidence to look for this source here.

- page 10 line 9: Why did you choose Sep-May and not Dec-Feb? Shouldn't that give you a better contrast?

- page 10 line 15: I don't know about the fuels in Finland, but methanol, acetaldehyde and acetone are usually not ingredients of gasoline. In many places, especially the USA and Brazil, gasoline contains a lot of ethanol, but usually no other oxygenated VOCs.

- page 10 lines 25-28: It is probably correct that in the road sector most aromatics are from traffic, but in general toluene and to a lesser extend C2-aromatics have large non traffic related sources such as solvents, paints and paint thinners. This should be mentioned here.

- page 11 line 9: The total traffic related flux of aromatics is calculated here and with this the fraction of the traffic to total aromatics flux can be estimated. As I mentioned earlier, this would be a very important result. Is the error of 1.2+/-0.2 g/m2/yr correct? Looking at the error estimate in the equation two lines above, this seems low?

- page 11 line 15: This is the only time Figure 9 is mentioned in this section of the text and is only briefly mentioned later on. Either this figure needs to be explained better or deleted.

- page 11 lines 20-28: It is clear that even in the road sector other sources besides traffic are strongly contributing to the VOC emissions, which can be seen in all the low correlation coefficients given in this paragraph. This would be a good place to mention the difficulty in the source apportionment again.

- page 11 lines 29-34: Here I am wondering again, if cycloalkanes are contributing to the signal on mass 69.

- page 12 line 21: The isoprene emissions, at least for the road and vegetation sectors, are of the same magnitude, but the CO2 emissions look very different. There is a very clear signature of CO2 uptake in the vegetation sector, and therefore one would expect

to have similar CO2 uptake in the road sector and as a result the anthropogenic CO2 flux is underestimated. Is this effect taken into account in this manuscript? Can this be used to estimate the flux and be compared to the Jaervi et al 2012 paper?

- page 13 lines 14-15 and page 13 lines 26-27: It is not clear to me, how these contributions are actually calculated, the biogenic monoterpene contribution and the non-traffic related contributions of anthropogenic VOCs. As I mentioned earlier, this is one of the more important results and it should be explained in detail how these contributions or "source attribution" is calculated.

- page 14 lines 1-8: I agree that acetonitrile from the built sector is likely from solvent use in the chemistry buildings, but again domestic burning cannot be well characterized using acetonitrile due to the low emission rate from domestic burning.

- Figure 2: please label the sectors and add typical footprints.

- Figure 3, 6 and 9: a label would make the figures much easier to look at.

- Figure 7: It would be worth showing the other sectors for comparison.

Technical Comment: There are so many small grammatical errors everywhere in the manuscript, mainly missing articles and prepositions, that I can not list them all here. I would suggest an additional proofreading.

---

## Referee Comment (RC2) · Anonymous Referee #1 · 11 Apr 2016

Rantala et al present long term flux measurements of VOCs, CO2 and periodically CO from an urban background site in Northern Finland. This represents the first such data set from a city in the northern latitudes and is therefore of interest as it builds upon our still very sparse collection of urban VOC flux data sets. I am therefore keen to see this work published, but I have some reservations about the methods used and in particular the simplicity of the division of the footprint into road, built and vegetation sectors. The authors must address these points before I can recommend publication.

Main Comments:

The authors segregate their measured fluxes into three distinct sectors (built, road and vegetation; defined in figure 1) and try to establish differences between the emission

rates observed in each. I think this is a worthwhile exercise as it goes beyond what has been published in previous urban VOC flux studies. However, in order to do this properly I would expect a much more detailed analysis of the flux footprint that allows the footprint contributions to be mapped to specific areas surrounding the tower e.g. major roads, buildings and vegetation. I have seen this type of analysis applied to fluxes measured over agricultural land (Neftel et al, 2008) and also urban areas (Helfter et al, 2011) and feel this might offer more meaningful results than your current approach of segregating sectors on the basis of wind direction (based on a study by Vesala et al. (2008)) which appears overly simplistic. For example, on occasions when the wind comes from the south the flux footprint would encompass both the road (i.e. the major highway) and vegetation sectors. Perhaps it is uncommon for the wind to come from the boundaries between sectors, but this is information is not included. How do the authors treat such periods where the footprint is likely to span two sectors? It would be very useful if the authors could supply wind roses for the different measurement periods in a supplementary information section so we can judge for ourselves whether this is an issue or not.

On page 6, line 27 the authors state "Other quality controlling, such as filtering flux data with flux detection limits or with stationarity criteria was not performed because applying these methods for the noisy DEC data would potentially bring other uncertainty sources". Could you please elaborate on this and define what you mean by "other uncertainty sources"? My interpretation is that you did not want to remove individual fluxes that fell below the limit of detection because your averaged fluxes would then be biased high. I would agree with this, but not filtering the raw data for data below the limit of detection means you subsequently need to convince us that your averaged fluxes are significantly different from zero. From page 5, line 20 we already know that the average of the data sets between calibrations are significant, but the same assurance is needed when you average the data for your various analyses e.g. by time of day. For example, in figure 3 (m/z 42), the red and blue traces do not look significantly different from zero to me. I would recommend calculating an averaged limit of detection (which

others have done, see Valach et al. 2015) for each of your analyses so we know for sure. This does not necessarily need to be added to the plots in the main manuscript but should certainly be shown in the SI.

The method used to calculate time-lags is clearly critical to determining the flux. A recent publication by Langford et al. (2015) demonstrated that significant bias (both positive and negative) can be introduced to noisy eddy covariance data when methods are used that search for a maximum in a cross-covariance function. They also suggest that the problem is exacerbated at high measurement points and when sampling through long inlet lines and especially for disjunct data which has poorer statistics and hence a higher random error. Your data set would appear to fit into this higher risk category and therefore I think it is important for you to demonstrate that your data are not affected by this bias. I appreciate that you suggest the potential bias is minimised through the use of a relatively small lag-time window and the use of the smoothed cross-covariance but depending on the signal-to-noise ratio of your data a significant bias could remain. This is important to know since you state in at least two sections that some of your fluxes were very close to the detection limit (Page 10, line 6 and Page 11, line 11). Given the length of your data set, recalculating the fluxes using a prescribed lag time is perhaps unrealistic, but it would certainly be interesting to see how the different time-lag methods compare over a shorter period of a few weeks and to see the flux distributions in the supplementary information. Such an analysis would give us further confidence in the fluxes you present. Related to this, on Page 5, line 18 please could you give more details on the method of smoothing you applied? Was this a running mean? How many data points were used for the average?

The method section 2.2 seems a little muddled and could do with restructuring and there is some important information missing. You start by introducing the DEC equation, but then immediately follow up with a discussion of high frequency loss corrections. It would make more sense to me for you to follow the equation with an outline of your flux calculation procedure. For example, you should mention at this point what the

length of the averaging period was, what the typical value of n was, what the duty cycle length was, what the typical time-lag was and how you calculated it etc. Once you have fully outlined how you calculated the fluxes you can then start your discussion of the flux corrections and QA/QC procedures you applied. Most of this information is there, it's just a case of restructuring in a more logical order.

Page 10, line 21. The emission potentials are not shown in figure 10. Figure 10 shows the regression between measured and modelled isoprene/furan fluxes from which the emission potential can be derived. In the text you need to make it more clear how you derived the emission potentials from figure 10, unless you are familiar with this type of analysis it is not obvious. In deriving the emission potentials did you set the intercept to equal zero? This information should be included.

I like the fact you have calculated isoprene emission potentials for urban vegetation, but in their current format I don't think they are particularly useful. Strictly speaking the G93 algorithm is used for leaf-level emission potentials on a mass per gram of dry leaf basis. While it can be used to derive area based emission potentials as you have done, the values are not likely to be compatible with the more recent BVOC emission models such as MEGAN that use area based emission factors, in part because these newer algorithms use a different set of standard conditions. In order to maximize the usefulness of these results I would suggest also converting your area based emission potentials to leaf-level potentials (ng g-1 s-1) by first estimating the foliar density for your flux footprint. Estimating the foliar density will of course introduce additional uncertainty and this should be factored in to your presented emission potentials. These values could then be compared to the standard urban isoprene emission potentials used in Guenther et al., (1995) and to those derived for other European cities (Valach et al. 2015).

Minor Corrections:

In the main text you discuss the fluxes and concentrations using the specific names

of the compounds measured, whereas you refer to the measured m/z ratio in your figures. As you have spent time in Section 2.2.2 identifying the m/z ratios I would suggest harmonising the figures with the text and using the compound(s) names.

Please can you clarify why you separate your data into Jun-Aug and Sep-May? While this isolates the warmest summer months, autumn, spring and winter are all wrapped together. With such an extensive set of measurements could you not have looked at the variation of VOC and $CO_2$ fluxes at a much finer temporal resolution (e.g. monthly... or at least by season) and compared with monthly variations in traffic and temperature? This would be very interesting as none of the previous urban VOC flux work published have shown monthly variations across a full year.

Page 2, line 4: suggest you change to "...have generally major effects on the chemistry of the atmosphere"

Page 2, line 9: change to: "...conducted in the UK where winters are relatively mild."

Page 2, line 13: please add the reference to which you are referring to.

Page 2, line 21. The climate zone descriptions given in Stewart and Oke are very brief so I would suggest adding a line to describe the characteristics of climate zone 6 so the reader doesn't have to look it up.

Page 3, line 2, change "blew" to "was"

Page 3, line 8, please change to "For the rest of the time..."

Page 3 line 14. Please add somewhere to this paragraph the Reynold's number for the two flow regimes used

Page 3, line 22. Please add the ± uncertainty of the Apel-Reimer gas standard used for calibration

Page 5, line 22. Please can you define what you mean by "...its flux values were defined to be insignificant". Does this mean the data were set to zero or rejected? If it

was the latter did you use gap-filling?

Page 6, line 17. Can you infer the low frequency flux losses from the co-spectral analysis applied to your CO2 fluxes?

Page 9, line 23. The measured CO/CO2 flux ratios could be further compared to those measured above London by Harrison et al. (2012).

Page 13, line 9. The monoterpene fluxes in figure 3 don't look any more or less scattered then any of your other diurnal cycles. Please rephrase this sentence to better reflect the data shown or remove.

Page 13, line 20. I think it's worth adding a line here to make it clear that you are using the intercept as a measure of the non-traffic related emissions.

Page 13, line 26. Again, please be clear about how you arrived at this estimate.

Page 13, line 34. Please change to "Nevertheless, the contribution from non-biogenic isoprene+furan emissions. . .."

Page 14, line 29. I would presume the ambient temperature also has a large effect on VOC emission rates? Was the ambient temperature higher in Mexico compared to London and might this have resulted in larger evaporative emissions? If so, I wonder if temperature can be incorporated into figure 11 in some way or mentioned in your discussion.

Page 29: Figure 1, please add the zero line for temperature.

Page 31: Figure 3 please add the y axis zero line to each plot

Page 33: Figure 5. I would recommend changing the blue circles to open circles. I would also expect to see error bars and a zero line shown on the y axis.

Page 34: Figure 6. I was interested to see that the CO fluxes are zero at night time but the CO2 flux is still showing emission. Can you provide some comment on this?

Secondly, could you also provide some further comment as to why the two peaks in CO flux do not correspond temporally with the peaks in $CO_2$ and traffic counts? It would be interesting to see how the ratio of the two change throughout the day. In addition please add the zero lines to the CO and $CO_2$ plots.

Page 35: Figure 7. Please add the zero lines.

Page 36: Figure 8. Please add the zero lines.

Page 38: Figure 10. Please add the zero lines.

References:

Guenther, A., Hewitt, C. N., Erickson, D., Fall, R., Geron, C., Graedel, T., Harley, P., Klinger, L., Lerdau, M., McKay, W. A., Pierce, T., Scholes, B., Steinbrecher, R., Tallamra, R., Taylor, J., and Zimmerman, P.: A global model of natural volatile organic compound emissions, J. Geophys. Res., 100, 8873–8892, doi:10.1029/94JD02950, 1995.

Harrison, R. M., Dall'Osto, M., Beddows, D. C. S., Thorpe, A. J., Bloss, W. J., Allan, J. D., Coe, H., Dorsey, J. R., Gallagher, M., Martin, C., Whitehead, J., Williams, P. I., Jones, R. L., Langridge, J. M., Benton, A. K., Ball, S. M., Langford, B., Hewitt, C. N., Davison, B., Martin, D., Petersson, K. F., Henshaw, S. J., White, I. R., Shallcross, D. E., Barlow, J. F., Dunbar, T., Davies, F., Nemitz, E., Phillips, G. J., Helfter, C., Di Marco, C. F., and Smith, S.: Atmospheric chemistry and physics in the atmosphere of a developed megacity (London): an overview of the REPARTEE experiment and its conclusions, Atmos. Chem. Phys., 12, 3065-3114, doi:10.5194/acp-12-3065-2012, 2012.

Helfter, C., Famulari, D., Phillips, G. J., Barlow, J. F., Wood, C. R., Grimmond, C. S. B., and Nemitz, E.: Controls of carbon dioxide concentrations and fluxes above central London, Atmos. Chem. Phys., 11, 1913–1928, doi: 10.5194/acp-11-1913-2011, 2011

Langford, B., Acton, W., Ammann, C., Valach, A., and Nemitz, E.: Eddy-covariance

data with low signal-to-noise ratio: time-lag determination, uncertainties and limit of detection, Atmos. Meas. Tech., 8, 4197-4213, doi:10.5194/amt-8-4197-2015, 2015.

Neftel, A., Spirig, C., and Ammann, C.: Application and test of a simple tool for operational footprint evaluations, Environ. Pollut., 152, 644–652, 2008.

Stewart, I. D. and Oke, T. R.: Local climate zones for urban temperature studies, Bulletin of the American Meteorological Society, 93, 1879-1900, 2012.

Valach, A. C., Langford, B., Nemitz, E., MacKenzie, A. R., and Hewitt, C. N.: Seasonal and diurnal trends in concentrations and fluxes of volatile organic compounds in central London, Atmos. Chem. Phys., 15, 7777-7796, doi:10.5194/acp-15-7777-2015, 2015.

Vesala, T., Jarvi, L., Launiainen, S., Sogachev, A., Rannik, U., Mammarella, I., Siivola, E., Keronen, P., Rinne, J., Riikonen, A., and Nikinmaa, E.: Surface-atmosphere interactions over complex urban terrain in Helsinki, Finland, Tellus B, 60, 188–199, 2008.

---

## Editor Comment (EC1) · T. Karl (Editor) · 12 Apr 2016

T. Karl (Editor)

thomas.karl@uibk.ac.at

Looking at the map provided in the figure 2, the site seems to be particularly influenced by green vegetation. Even the sector identified as buildings is comprised of a fairly large fraction of vegetation, rather unique compared to the urban UK sites the authors compare their measurements with. Consistently this sector does not show a significantly different pattern of m/z 69 emissions. Perhaps a better description of this sector would be to call it ' urban residential sector with vegetation'.

BTEX emissions: The sector identified as road boarders to what it seems like an industrial complex (for example: at a distance of about 300-400 m a smoke stack is evident on google earth). It is argued that this sector is primarily influenced by road traffic. The

influence of additional BTEX sources in this sector (other than traffic) could perhaps be obtained by explicitly comparing toluene to benzene fluxes during rush hour peaks with other periods. The upper limit of traffic related emission ratios should be close to 2 (1.9) based on the emission factor database for the average European fleet. The authors compare their measurements to other cities. In this context it is noted that Mexico City seems to be a special place with respect to many of the measured VOC fluxes. For example toluene measurements by Velasco et al., 2009, were thought to be influenced by local application of resin surrounding the flux tower resulting in toluene / benzene flux ratios of about 8-10. Measurements by Karl et al., 2009, reported a city wide average ratio of about 3.2 for Mexico City and concluded that about 60-70% of toluene could be due to evaporative emissions. Figure 11: It is noted that a correlation of fluxes between some compounds (such as toluene) and $CO_2$ needs to be discussed with caution. For example most of traffic related toluene emissions are evaporative and not produced by the ICE - thus not intrinsically linked to $CO_2$ tailpipe emissions. This is fundamentally different for benzene emissions for example, which are much more closely related to tail-pipe emissions.

---

## Author Comment (AC1) · 31 May 2016

We thank both Referees for the excellent reviews and good suggestions. We have adapted and answered all the comments and the revised manuscript has been significantly improved. Below detailed answers to the comments can be found. The referee comments are bolded whereas our replies are written in a normal text.

**REFEREE #1**

**Rantala et al present long term flux measurements of VOCs, CO2 and periodically CO from an urban background site in Northern Finland. This represents the first such data set from a city in the northern latitudes and is therefore of**

**interest as it builds upon our still very sparse collection of urban VOC flux data sets. I am therefore keen to see this work published, but I have some reservations about the methods used and in particular the simplicity of the division of the footprint into road, built and vegetation sectors. The authors must address these points before I can recommend publication. Main Comments: The authors segregate their measured fluxes into three distinct sectors (built, road and vegetation; defined in figure 1) and try to establish differences between the emission rates observed in each. I think this is a worthwhile exercise as it goes beyond what has been published in previous urban VOC flux studies. However, in order to do this properly I would expect a much more detailed analysis of the flux footprint that allows the footprint contributions to be mapped to specific areas surrounding the tower e.g. major roads, buildings and vegetation. I have seen this type of analysis applied to fluxes measured over agricultural land (Neftel et al, 2008) and also urban areas (Helfter et al, 2011) and feel this might offer more meaningful results than your current approach of segregating sectors on the basis of wind direction (based on a study by Vesala et al. (2008)) which appears overly simplistic. For example, on occasions when the wind comes from the south the flux footprint would encompass both the road (i.e. the major highway) and vegetation sectors. Perhaps it is uncommon for the wind to come from the boundaries between sectors, but this is information is not included. How do the authors treat such periods where the footprint is likely to span two sectors? It would be very useful if the authors could supply wind roses for the different measurement periods in a supplementary information section so we can judge for ourselves whether this is an issue or not.**

We agree that calculating proper footprint estimates would allow a much more detailed source analysis. However, the parameterized analytical footprint models (such as Kljun or Kormann and Meixner) commonly used in EC studies do not function well in heterogeneous surroundings and therefore dividing the source area into detailed patches of different land uses is not meaningful. This is particularly true at the Kumpula site where

calculation of flux footprints using a bit more sophisticated model in neutral conditions has shown somewhat different pattern (Vesala et al. 2008) when compared to the simple elliptic footprints commonly obtained from the analytical models. Unfortunately the problem with this model (and commonly with more complex models) is that only neutral conditions can be calculated. Nevertheless, we added the cumulative 80% footprint to the Fig. 2 calculated using the Korman and Meixner model to give some indication about the source area of the measurements.

We also think that flux footprints can be spanning over two sectors and thus we now filtered the data based on the footprint estimates. A measured flux value was defined to be, for example, from the road sector if maximum 30% of the 80% flux footprint area covered other than the road sector. Thus, periods when wind blew close to a sector border, were rejected from further analysis. The total rejection rate was around 30%. On the other hand, disregarding data decreased also statistical significance, especially in the case of the built sector. Nevertheless, we decided to do the division into different sectors based on the footprint estimated source areas to avoid the problems pointed out by Referee #1. All the Figures and Tables were changed correspondingly and text related to the method was added on P10, L24-28.

We added the median fluxes from different wind directions (20° bins) to the supplementary material.

**On page 6, line 27 the authors state "Other quality controlling, such as filtering flux data with flux detection limits or with stationarity criteria was not performed because applying these methods for the noisy DEC data would potentially bring other uncertainty sources". Could you please elaborate on this and define what you mean by "other uncertainty sources"? My interpretation is that you did not want to remove individual fluxes that fell below the limit of detection because your averaged fluxes would then be biased high. I would agree with this, but not filtering the raw data for data below the limit of detection means you subsequently need to convince us that your averaged fluxes are significantly different**

**from zero. From page 5, line 20 we already know that the average of the data sets between calibrations are significant, but the same assurance is needed when you average the data for your various analyses e.g. by time of day. For example, in figure 3 (m/z 42), the red and blue traces do not look significantly different from zero to me. I would recommend calculating an averaged limit of detection (which others have done, see Valach et al. 2015) for each of your analyses so we know for sure. This does not necessarily need to be added to the plots in the main manuscript but should certainly be shown in the SI.**

We mean that removing flux values based on, for example, detection limits, can easily bias average values so the interpretation of the Referee #1 is correct. In our opinion, somewhat noisy DEC data should be filtered using only independent data, such as friction velocities. We would not like to do any filtering based on the DEC data itself because we are not sure if this can create systematic error sources. Only exception is that measurement periods with lot of spikes were of course disregarded from further analysis. This was clarified in the revised manuscript.

We agree with the referee that acetonitrile flux does not differ statistically significantly from zero except from the built sector. This is also mention in the original manuscript (page 12, line 7–8; page 14, line 1). Furthermore, we admit that detection limits would be very useful information for the reader, thus we added mean flux detection limits ($\overline{\text{LoD}}$) to Table 4 (revised manuscript) for each season. Individual LoDs were defined to be $1.96 \times \sigma_{ccf}$, where $\sigma_{ccf}$ is the standard deviation of the cross covariance function tails (see Taipale et al. 2010). The mean $\overline{\text{LoD}}$ was then calculated using a formula $\overline{\text{LoD}} = 1/N \sum \text{LoD}^2$ as discussed in Valach et al. (2015). Single flux values were of course under detection limits more often but the average fluxes were not.

**The method used to calculate time-lags is clearly critical to determining the flux. A recent publication by Langford et al. (2015) demonstrated that significant bias (both positive and negative) can be introduced to noisy eddy covariance data when methods are used that search for a maximum in a cross-covariance func-**

**tion. They also suggest that the problem is exacerbated at high measurement points and when sampling through long inlet lines and especially for disjunct data which has poorer statistics and hence a higher random error. Your data set would appear to fit into this higher risk category and therefore I think it is important for you to demonstrate that your data are not affected by this bias. I appreciate that you suggest the potential bias is minimised through the use of a relatively small lag-time window and the use of the smoothed cross-covariance but depending on the signal-to-noise ratio of your data a significant bias could remain. This is important to know since you state in at least two sections that some of your fluxes were very close to the detection limit (Page 10, line 6 and Page 11, line 11). Given the length of your data set, recalculating the fluxes using a prescribed lag time is perhaps unrealistic, but it would certainly be interesting to see how the different time-lag methods compare over a shorter period of a few weeks and to see the flux distributions in the supplementary information. Such an analysis would give us further confidence in the fluxes you present. Related to this, on Page 5, line 18 please could you give more details on the method of smoothing you applied? Was this a running mean? How many data points were used for the average?**

Yes, determining the lag-times properly is one of the most important tasks in flux calculations. In our case, the cross covariance functions were usually quite noisy due to low fluxes and limited amount of data points for each 45-min-period. Thus, single flux values were usually close to a detection limit or below it. As the Referee mentions, this behaviour may lead to a strong bias if the maximum method with a wide time-time window is used for searching the lag-times.

However, we tried to minimize this behaviour by determining first a mean lag-time for each compound, and then seeking the individual lag-times using a short $\pm2.5$ s lag-time window and smoothed cross covariance functions. On the other hand, according to Taipale et al. (2010), a constant lag time should be avoided as well because then

fluxes are then easily underestimated. Langford et al. (2015) mentions that the problem can be partly avoided by controlling the flow rate, heating the inlet line and recording wind and concentration data to a same computer. However, our flow rate was not controlled via a mass flow controller and the data was also recorded to two computers. Thus, small variations in lag-times can be expected and using a constant lag-time would probably underestimate the fluxes. On the other hand, we did not want use wider lag-time window because then the mirroring effect would become more visible.

We added to the supplementary material the flux distributions for each compound. The distributions were calculated using a constant (mean) lag-time, and using a lag-time window of $\pm 2.5$ s around the mean (this study). The period was May 21 – June 4 2013. The distributions were quite equal for many compounds. The average fluxes with a constant lag-time were typically lower (up to 30%) but we think that this is caused by the fact the actual lag-time does not stay totally constant. Of course, random variation affects also the results as only 147 data points were used in the study. After that said, we admit that when dealing with fluxes close to the detection limit, values can be somewhat biased. We added more discussion about the topic to the revised manuscript (Section 2.2.1).

The smoothing applied in the study was based on a running mean with an averaging window of $\pm$ 2.4 s, i.e. 49 data points. We added this information to the text (Section 2.2.1).

**The method section 2.2 seems a little muddled and could do with restructuring and there is some important information missing. You start by introducing the DEC equation, but then immediately follow up with a discussion of high frequency loss corrections. It would make more sense to me for you to follow the equation with an outline of your flux calculation procedure. For example, you should mention at this point what the length of the averaging period was, what the typical value of n was, what the duty cycle length was, what the typical time-lag was and how you calculated it etc. Once you have fully outlined how you**

**calculated the fluxes you can then start your discussion of the flux corrections and QA/QC procedures you applied. Most of this information is there, it's just a case of restructuring in a more logical order.**

We thank for the suggestion and re-organized Section 2.2.1. First, we present the DEC equation and the measurements with the PTR-MS. However, basic details about the measurements are already discussed in Section 2.1.1. High frequency corrections and potential flux uncertainties are discussed at the end of the section.

**Page 10, line 21. The emission potentials are not shown in figure 10. Figure 10 shows the regression between measured and modelled isoprene/furan fluxes from which the emission potential can be derived. In the text you need to make it more clear how you derived the emission potentials from figure 10, unless you are familiar with this type of analysis it is not obvious. In deriving the emission potentials did you set the intercept to equal zero? This information should be included. I like the fact you have calculated isoprene emission potentials for urban vegetation, but in their current format I don't think they are particularly useful. Strictly speaking the G93 algorithm is used for leaf-level emission potentials on a mass per gram of dry leaf basis. While it can be used to derive area based emission potentials as you have done, the values are not likely to be compatible with the more recent BVOC emission models such as MEGAN that use area based emission factors, in part because these newer algorithms use a different set of standard conditions. In order to maximize the usefulness of these results I would suggest also converting your area based emission potentials to leaf-level potentials (ng g-1 s-1) by first estimating the foliar density for your flux footprint. Estimating the foliar density will of course introduce additional uncertainty and this should be factored in to your presented emission potentials. These values could then be compared to the standard urban isoprene emission potentials used in Guenther et al., (1995) and to those derived for other European cities (Valach et al. 2015).**

The emission potentials were calculated using the G93 algorithm, i.e. the parameter $E_0$ was fitted to the data. Intercept was defined to be zero. Of course, other than biogenic isoprene emissions contributed also to the flux at m/z 69. However, these emissions were estimated to minor compared with the biogenic ones, thus, no intercept etc. was allowed when the emission potentials were determined. We clarified this in the text (Section 3.2.2).

We agree that the emission potentials are not very useful from the modelling point of view as the vegetation coverage is heterogeneous. However, our purpose was to show that the flux at m/z 69 consists mostly of biogenic isoprene as the results agree well with the G93 algorithm. In addition, we wanted to point out that the (normalized) isoprene emissions are quite uniform from all wind directions. Indeed, the results would be more useful is they were scaled to leaf-level. In this approach, they could be also compared with other urban studies. Unfortunately, we think that estimating the dry leaf masses would be very inaccurate because tree species diversity around the site is large, partly due to the University botanical garden. Therefore, we would avoid to do such analysis. We also think that more complicated MEGAN algorithm would bring no benefit for our purposes.

As a conclusion, we left Fig. 12 in the revised manuscript but removed the wind direction separation as Referee #2 suggested. The zero lines were also added. However, we removed the Table 5 and also discussed in the text that the emission potentials cannot be compared with the other studies due to the problems pointed out by the Referee #1 (Section 3.2.2).

**Minor Corrections**

**In the main text you discuss the fluxes and concentrations using the specific names of the compounds measured, whereas you refer to the measured m/z ratio in your figures. As you have spent time in Section 2.2.2 identifying the m/z ratios I would suggest harmonising the figures with the text and using the**

**compound(s) names.**

We agree with this. In the revised manuscript, compound names are used in all figures instead of mass-to-charge ratios. We used the actual names also in Tables 3–4 in the revised manuscript. However, naming in Table A1 was not changed because otherwise the table would have become too large.

**Please can you clarify why you separate your data into Jun-Aug and Sep-May? While this isolates the warmest summer months, autumn, spring and winter are all wrapped together. With such an extensive set of measurements could you not have looked at the variation of VOC and CO2 fluxes at a much finer temporal resolution (e.g. monthly... or at least by season) and compared with monthly variations in traffic and temperature? This would be very interesting as none of the previous urban VOC flux work published have shown monthly variations across a full year.**

The separation was done because we wanted to see if the warmest season differs from other months. Traffic rates were also lowest during summer while they stayed otherwise quite constant. We agree the conditions vary a lot between September and May, thus the separation was not perfect from that point of view. We decided to present the seasonal cycles in the revised manuscript because the flux data coverage was not good enough to present data in monthly basis. The results and discussion was changed accordingly. See also response for the Referee #2.

**Page 2, line 4: suggest you change to "...have generally major effects on the chemistry of the atmosphere"**

Changed.

**Page 2, line 9: change to: "...conducted in the UK where winters are relatively mild."**

Changed.

**Page 2, line 13: please add the reference to which you are referring to.**

We added the reference (Langford et al., 2010). In addition to that, Harrison et al. (2012) studied also relationships between CO and VOCs, thus, that reference was also included to the introduction.

**Page 2, line 21. The climate zone descriptions given in Stewart and Oke are very brief so I would suggest adding a line to describe the characteristics of climate zone 6 so the reader doesn't have to look it up.**

The site is classified as local climate zone, which corresponds to "open low-rise" (see Stewart and Oke, 2012) with detached buildings and scattered trees and abundant vegetation. We described the climate zone better in the revised manuscript (Section 2.1).

**Page 3, line 2, change "blew" to "was"**

Fixed.

**Page 3, line 8, please change to "For the rest of the time..."**

Changed.

**Page 3 line 14. Please add somewhere to this paragraph the Reynold's number for the two flow regimes used**

We added the Reynolds numbers to the paragraph.

**Page 3, line 22. Please add the $\pm$ uncertainty of the Apel-Reimer gas standard used for calibration**

The uncertainty of the standard gas ($\pm 5\%$) was added to the text.

**Page 5, line 22. Please can you define what you mean by "...its flux values were defined to be insignificant". Does this mean the data were set to zero or rejected? If it was the latter did you use gap-filling?**

We tried to say that those mass-to-charge ratios with no significant peak values at all were rejected from the further study. This concerns mass-to-charge ratios 31, 89 and 103. We clarified this in the text.

Generally, no gap-filling was used because the procedure would be very complicated above the heterogeneous terrain with multiple sources.

**Page 6, line 17. Can you infer the low frequency flux losses from the co-spectral analysis applied to your CO2 fluxes?**

The corrections were $< 3\%$ and we mentioned this in the text.

**Page 9, line 23. The measured CO/CO2 flux ratios could be further compared to those measured above London by Harrison et al. (2012).**

We thank for the reference. Harrison et al. (2012) found a CO/CO2-ratio of 0.32–0.55% whereas in our study the ratio was 0.34%. We added this comparison to the text.

**Page 13, line 9. The monoterpene fluxes in figure 3 don't look any more or less scattered then any of your other diurnal cycles. Please rephrase this sentence to better reflect the data shown or remove.**

We agree with this statement and removed the sentence from the text.

**Page 13, line 20. I think it's worth adding a line here to make it clear that you are using the intercept as a measure of the non-traffic related emissions.**

We clarified the section to point out that the intercept was used as a measure of the non-traffic related emissions.

**Page 13, line 26. Again, please be clear about how you arrived at this estimate.**

We admit that the procedure was not well described. The estimate was rough and was based on the intercepts of the linear fits between the OVOC fluxes and the traffic rates. The intercepts were compared with the measured average fluxes. Considering

relatively high uncertainty estimates, we concluded that the emissions from traffic and other anthropogenic sources were around the same.

However, we improved the source identification as was asked by Referee #2, and the procedure is currently better explained in the revised manuscript. We also tried to give more accurate value with uncertainty estimates. We concluded that the traffic can explain $65 \pm 25$% of the measured OVOC flux at the site (Table 6).

**Page 13, line 34. Please change to "Nevertheless, the contribution from non-biogenic isoprene+furan emissions...."**

Changed.

**Page 14, line 29. I would presume the ambient temperature also has a large effect on VOC emission rates? Was the ambient temperature higher in Mexico compared to London and might this have resulted in larger evaporative emissions? If so, I wonder if temperature can be incorporated into figure 11 in some way or mentioned in your discussion.**

We agree with this statement. The average temperature was around $13°C$ ($12.2°C$ at the 95 m tall tower) in London during the campaign whereas in Mexico city, the ambient temperature was somewhat higher, varying diurnally between 10 and $25°C$ (Fast et al., 2007). For example evaporative solvent emissions might increase as a function of the ambient temperature. We added discussion about the topic to the text (Section 3.3).

**Page 29: Figure 1, please add the zero line for temperature.**

The zero line was added for temperature.

**Page 31: Figure 3 please add the y axis zero line to each plot**

The zero lines were added to each plot.

**Page 33: Figure 5. I would recommend changing the blue circles to open circles.**

**I would also expect to see error bars and a zero line shown on the y axis.**

The blue circles were changed to open circles as suggested. Error bars and a zero line were also added. See also the response for Referee #1.

**Page 34: Figure 6. I was interested to see that the CO fluxes are zero at night time but the CO2 flux is still showing emission. Can you provide some comment on this? Secondly, could you also provide some further comment as to why the two peaks in CO flux do not correspond temporally with the peaks in CO2 and traffic counts? It would be interesting to see how the ratio of the two change throughout the day. In addition please add the zero lines to the CO and CO2 plots.**

The non-negative nocturnal $CO_2$ fluxes origin mainly from the soil and vegetation respiration from the vegetation near the station (see Järvi et al., 2012). We mentioned this in the revised manuscript.

CO-flux is also peaking during the rush hours (see upper quartiles in Fig. 6, original manuscript) but interestingly the highest median fluxes were observed couple of hours later. As the amount of CO-flux data was quite limited, this is might be also coincidence. Some CO emissions could originate also from a residential building area behind the road. We studied the $CO/CO_2$ relations more in the revised manuscript (Section 3.2).

The zero lines were added to the CO and $CO_2$ plots.

**Page 35: Figure 7. Please add the zero lines.**

**Page 36: Figure 8. Please add the zero lines.**

**Page 38: Figure 10. Please add the zero lines.**

The zero lines were added to Figs. 7, 8 and 10 (Figs. 10–12 in the revised manuscript).

References:

[Figure]

Fast, J. D., de Foy, B., Acevedo Rosas, F., Caetano, E., Carmichael, G., Emmons, L., McKenna, D., Mena, M., Skamarock, W., Tie, X., Coulter, R. L., Barnard, J. C., Wiedinmyer, C., and Madronich, S.: A meteorological overview of the MILAGRO field campaigns, Atmospheric Chemistry and Physics, 7, 2007.

**Guenther, A., Hewitt, C. N., Erickson, D., Fall, R., Geron, C., Graedel, T., Harley, P., Klinger, L., Lerdau, M., McKay, W. A., Pierce, T., Scholes, B., Steinbrecher, R., Tallamra, R., Taylor, J., and Zimmerman, P.: A global model of natural volatile organic compound emissions, J. Geophys. Res., 100, 8873–8892, doi:10.1029/94JD02950, 1995.**

**Harrison, R. M., Dall'Osto, M., Beddows, D. C. S., Thorpe, A. J., Bloss, W. J., Allan, J. D., Coe, H., Dorsey, J. R., Gallagher, M., Martin, C., Whitehead, J., Williams, P.I., Jones, R. L., Langridge, J. M., Benton, A. K., Ball, S. M., Langford, B., Hewitt, C. N., Davison, B., Martin, D., Petersson, K. F., Henshaw, S. J., White, I. R., Shallcross, D. E., Barlow, J. F., Dunbar, T., Davies, F., Nemitz, E., Phillips, G. J., Helfter, C., Di Marco, C. F., and Smith, S.: Atmospheric chemistry and physics in the atmosphere of a developed megacity (London): an overview of the REPARTEE experiment and its conclusions, Atmos. Chem. Phys., 12, 3065-3114, doi:10.5194/acp-12-3065-2012, 2012.**

**Helfter, C., Famulari, D., Phillips, G. J., Barlow, J. F., Wood, C. R., Grimmond, C. S.B., and Nemitz, E.: Controls of carbon dioxide concentrations and fluxes above central London, Atmos. Chem. Phys., 11, 1913–1928, doi: 10.5194/acp-11-1913-2011, 2011**

Langford, B., Nemitz, E., House, E., Phillips, G. J., Famulari, D., Davison, B., Hopkins, J. R., Lewis, A. C., and Hewitt, C. N.: Fluxes and concentrations of volatile organic compounds above central London, UK, Atmospheric Chemistry and Physics, 10, 627–645, 2010.

**Langford, B., Acton, W., Ammann, C., Valach, A., and Nemitz, E.: Eddy-**

covariance data with low signal-to-noise ratio: time-lag determination, uncertainties and limit of detection, Atmos. Meas. Tech., 8, 4197-4213, doi:10.5194/amt-8-4197-2015, 2015.

Kormann, R., & Meixner, F. X.: An analytical footprint model for non-neutral stratification, Boundary-Layer Meteorology, 99(2), 207-224, 2001.

**Neftel, A., Spirig, C., and Ammann, C.: Application and test of a simple tool for operational footprint evaluations, Environ. Pollut., 152, 644–652, 2008.**

**Stewart, I. D. and Oke, T. R.: Local climate zones for urban temperature studies, Bulletin of the American Meteorological Society, 93, 1879-1900, 2012.**

Taipale, R., Ruuskanen, T. M., and Rinne, J.: Lag time determination in DEC measurements with PTR-MS, Atmospheric Measurement Techniques, 3, 853–862, 2010.

**Valach, A. C., Langford, B., Nemitz, E., MacKenzie, A. R., and Hewitt, C. N.: Seasonal and diurnal trends in concentrations and fluxes of volatile organic compounds in central London, Atmos. Chem. Phys., 15, 7777-7796, doi:10.5194/acp-15-7777-2015, 2015.**

**Vesala, T., Jarvi, L., Launiainen, S., Sogachev, A., Rannik, U., Mammarella, I., Siivola, E., Keronen, P., Rinne, J., Riikonen, A., and Nikinmaa, E.: Surface-atmosphere interactions over complex urban terrain in Helsinki, Finland, Tellus B, 60, 188–199, 2008.**

**REFEREE #2**

**General Comments: This paper describes >1.5 year long-term flux measurements of VOC, CO and CO2 in the urban environment of Helsinki, Finland. As was shown in the paper and also expected for an urban environment, most VOCs have large traffic related emissions, but other anthropogenic sources are also important for some VOCs. In addition, during summer biogenic isoprene and monoterpene emissions and CO2 uptake are evident in the data. Overall VOC fluxes in the specific location of these measurements were rather small compared to other cities. Long-term flux measurements, especially in an urban area, have not been reported in the literature very often and therefore this dataset is very interesting and unique and I think a dataset like this is worth exploring and publishing, but the analysis presented here needs major improvements before it is acceptable. Major Issues: 1. The organization of the discussion section: I found this paper very hard to read, because of a constant mix of topics in the first part of the discussion. I would suggest to re-organize the chapters 3.1 and 3.2., before discussing the individual emission sources (traffic, biogenic, and others). The seasonal and diurnal cycles for all VOCs, CO and CO2 should be discussed in detail first then discuss individual sources. I would like to see an actual figure showing the annual cycle for VOC, CO and CO2 fluxes, although there might not be enough data for CO. Right now this important information is hidden in various figures and tables. For this discussion the data should not be separated into the three sectors. After describing these general trends in the fluxes, each emission source sector can be described: traffic, biogenic, and others; and for all of these CO and CO2 should be included and not be shown in a separate chapter.**

We thank for the good suggestion. We re-organized the manuscript so that first annual cycles (seasonal averages) of measured fluxes and concentrations are discussed (Section 3.1) and shown in Fig. 5 in the revised manuscript. Tables and figures were also modified accordingly. Even though the measurements were done between Jan

2013 – Sep 2014, some months, mainly January, February and October were underrepresented in the data sets (less than 100 data points) and no good data from November and December was recorded at all. This was caused by long measurement gaps (Fig. 1 in the original manuscript). We believe that such a small amounts of data would not represent well the monthly averages, especially when taking to account the effect of the wind direction. For example, the wind did not blew from road sector almost at all in October, leading to really small fluxes of benzene, toluene and $C_2$-benzenes. Therefore, we decided to present seasonal cycles in the revised manuscript.

Unfortunately, CO-fluxes were only measured between April and May 2014, thus no annual cycles for CO is shown. However, the average CO-fluxes are still shown in Table 5. We also added a diurnal cycle of CO concentrations to the revised manuscript (Fig. 6).

**It would also be important to add the mixing ratios to the annual and diurnal cycles. From the paper as is, it is not possible to understand, if this is a heavily polluted location or not. I would assume that in the cold winter months, when the boundary layer is really low, mixing ratios could get rather high.**

We agree with this statement. The original manuscript contained only a basic mixing ratio statistics from the summer and the other months, but we expanded Table 3 in the revised manuscript to cover all four seasons. In addition, we added 95% quantiles which represent higher end of the measured concentrations. Furthermore, we added the diurnal median cycle for each compound and expanded discussion in Section 3.1.

However, we would like to avoid of analysing the concentrations in more detail for two reasons: First of all, the manuscript can easily become too long and its focus blurred if lot of concentration related material is added. We think that VOC concentrations are mostly driven by horizontal advection, not by the local emissions. Thus, these two components are difficult to analyse together. Secondly, the VOC concentrations (excluding alcohols) measured in Helsinki are already analysed in detail by Hellén et

al. (2003), (2006) and (2012).

We think that the pollution episodes with high mixing ratios were quite rare due to several reasons. First of all, Helsinki is rather small city and pollution emission are generally low. Secondly, the city is located by the sea, thus, totally calm situation when emissions could accumulate near the ground are somewhat uncommon. However, concentration of many compounds peaked during morning rush hour, probably as a result of traffic peak and relatively shallow boundary layer (revised manuscript, Fig. 6).

**The separation of the data into the three wind direction sectors looks like a good approach, when looking at the map and the potential emissions from those sectors, but the VOC flux data (Figure 3, 4 and 5) are actually very similar for each sector. The only substantial difference was found for CO and CO2 in Figure 6 and for the weekday/weekend plot in Figure 9, although that is mainly due to the traffic counts between weekday and weekend. The wind sector separation complicates the discussion in many places, but doesn't really add any information, so I think Figures 3, 4, 5, 9 and 10 should be simplified by using all the data. In addition, throughout the manuscript it becomes clear that even in the road sector substantial non-traffic related emissions are evident and in all three wind direction sectors multiple sources contribute to the VOC emissions. This makes a quantitative analysis and separation of sources very difficult and this should be acknowledged clearly in the manuscript.**

We agree with the referee that the differences between the sectors are not as clear as for CO and $CO_2$. However, the sector separation has been used in former flux publications at the site (Vesala et al., 2008; Järvi et al., 2012, 2014). Thus, we would still like to keep the separation, at least from the point of the comparison to the previous studies. In addition, differences between the sectors were clear for acetonitrile and acetone, both having most significant source in the built sector. In addition, the fluxes from the road sector were significantly higher for methanol, acetaldehyde, isoprene, benzene and $C_2$-benzenes. Therefore, we partly disagree with the comment that no

differences between the sectors were found. However, we agree that diving the data to three sectors gives no additional information for Figs. 4, 5, 9 and 10. Thus, those figures were re-plotted without the separation in the revised manuscript (Figs. 4, 7, 8 and 12).

Based on a suggestion from Referee #1, we decided to do all calculations using a footprint analysis instead of the wind directions (see a response for Referee #1), thus minimizing the interaction between the sectors. As a result, the differences between the sectors are clearer in some cases.

We also think that the quantitative analysis of the sources is tricky and we clarified this in the text (Sections 3.2.1 and 3.3).

**For some VOCs an attempt for a quasi source apportionment was done in the paper. For example, on page 13 line 10-15, the monoterpene sources are summarized and biogenic contribution was assumed to be around 40%. This type of information is to me one of the most relevant results of this paper. Unfortunately this estimation of a source apportionment was only done for monoterpenes and OVOCs and it would be important to do this carefully for all the measured VOCs. If this is possible with the data, I would like to see something like a pie chart for each VOC or class of VOCs showing the traffic, biogenic and other anthropogenic contributions for summer and winter, which should then be presented as the main result of this paper.**

We agree that this kind of information would be very useful. We expanded the discussion about sources and made a Table (Table 6 in the revised manuscript) that shows different sources of OVOCs, terpenes and aromatic compounds in summer and winter. Estimating the contributions has its uncertainties and this was pointed out in the revised manuscript (Section 3.3).

**Specific comments: page 2 line 26: It would be really helpful to add the typical footprint to Figure 2. The discussion about the wind sector analysis would be**

**much easier to follow.**

We added the cumulative 80% footprint to the Figure 2. In addition, the picture was enlarged to cover the area of 2000 × 2000 m instead of 1200 × 1200 m. See also the response to Referee #1.

**page 3 line 7: I assume this is 0.5s per mass per measurement cycle of about 6s each?**

The measurement cycle was slightly higher, around 7 s, because switching between the masses took some time, as well as the basic measurements (m/z 21 and m/z 37). We clarified this in the text.

**page 3 line 8-9: Were there any other masses with significant signal or was most of the VOC signal captured by the masses used for the presented flux measurements. Please indicate other important masses.**

The additional masses were m/z 61, 71, 73, 75, 87, 99, 101, 113 and 117. Furthermore, some heavier masses than m/z 137 were measured but the data quality was really poor due to low sensitivity of the PTR-MS at higher masses. We think that the most important mass-to-charge ratios were already included to the flux measurement cycle.

**page 3 line 19-21: The PTR-MS instrument settings are described here, but the actual detection limits for the 0.5s measurements and the 20-30 min flux calculations are not given. Please add those, especially taking the issues with the instrument background measurements into account.**

The flux detection limits were added to the Table 4. See also the response for Referee #1. The detection limits of concentration measurement were added to the Table 2. We determined the detection limits from the zero air measurements of the calibrations, thus, the detection limits are shown only for the calibrated compounds. The determined values represent $1.96\sigma$ detection limits for individual 0.5 s measurements.

**page 3 line 24: What do you mean by "correct primary ion signal"? Is this mass discrimination corrected? Shouldn't the calibration be done at the same settings as the actual measurements and not with optimized SEM voltages?**

The calibrations were always done before a measurement period. Therefore, the SEM voltage was optimized (increased) before the calibration and the same value was used until the next calibration (and the measurement period). The sentence was reformulated.

**page 3 line 25: Does the zero air generator change the humidity? Background measurements at a different humidity can produce significant artifacts.**

No, we used the ambient air also for the zero air measurements and humidity should not have changed. The artifacts were observed for toluene only but the reason for this remained unknown. We clarified this in the text.

**page 3 line 31-32: How much does the uncertain zero air measurement add to the uncertainty, please be specific.**

This depends on the compound but the possible systematic errors were estimated to be small. The procedure affects individual hourly values but this kind of data is not presented in the manuscript. We included to the text a note that the effect was estimated to be negligible.

**page 4 line 9: Also for CO and CO2 measurements it would be good to add the precision and uncertainties.**

The random error and detection limit of CO were 0.23 $\mu$g m$^{-2}$s$^{-1}$ and 0.16 $\mu$g m$^{-2}$s$^{-1}$, respectively. The corresponding numbers for CO$_2$ were 0.05 $\mu$g m$^{-2}$s$^{-1}$ and 0.03 $\mu$g m$^{-2}$s$^{-1}$, respectively. We added the missing information to the manuscript.

**page 4 line 16 and page 5 line 11: Why did you use different averaging times for CO and CO2 with 30 min compared to the VOCs with 45 min?**

For VOCs, we used 45 min averaging period to include more data for the flux calculations. This is crucial issue from the point of finding correct lag times. CO and $CO_2$ were measured properly with 10 Hz frequency, thus, there was no need for using longer 45 min intervals. Based on previous studied at the site the optimal flux calculation time for $CO_2$ has been found to be 30-minutes.

**page 5 line 13: Was there a reason not to use something like a time server synchronization program?**

We were not able to setup such a synchronization. Therefore we needed to do the time synchronization afterwards.

**page 5 line 14: Was m37 the highest flux, higher than methanol? Is that why m37 was used for the time lag calculation?**

Yes, the first water cluster showed generally highest – or better to say most clear – cross covariance function peaks. However, the actual lag-times were calculated individually for each compound. The lag times of the water cluster were only used to handle the shift between the computer clocks (the anemometer and the PTR-MS).

**page 6 line 31: Are those data coverages for flux measurements or do those include the mixing ratio measurements?**

The data coverages are for flux measurements; mixing ratios have slightly higher coverages. We clarified this in the text.

**page 7 lines 8-14: There have been a few recent papers about oil and gas emissions using PTR-MS showing that m69 can also have a significant influence from cycloalkanes.**

We thank the referee for this statement. For example, part of the cyclohexane fragments to m/z 69. According to Hellén et al. (2006), the cyclohexane concentrations are around 0.10 ppbv in Helsinki during winter, thus, affecting partly also observed m/z 69 signal. We added this additional information to the text, Tables and Figures.

**page 7 line 19: Are those anthropogenic monoterpene likely from the sector "solvents and other products" or more traffic related?**

We do not know. Hellén et al. (2012) speculated that monoterpenes could originate also from traffic but the possible processes are unknown. We think that traffic could be the most obvious solution as no industrial areas etc. are located nearby the station but this is hard to say. Part of the monoterpene emissions could originate at least from glass cleaner liquids. We expanded the discussion about anthropogenic sources of monoterpenes (3.2.3).

**page 7 line 33: Mention here that acetonitrile is often used as a tracer for biomass burning.**

We expanded the sentence to cover this information.

**page 8 line 6: Often biogenic inventories do not represent urban environments well, please explain where you get the E0,synth values for the measurement location from and what your confidence in this value is.**

We agree with this statement. Generally, the emission potential values are not necessary very useful in urban environment without having additional information about vegetation cover. Our purpose was only to show that the G93 algorithm works well for m/z 69 fluxes, thus, biogenic isoprene emissions have probably a major contribution to the measured flux at m/z 69 during summer. In addition, we wanted to show that (normalized) isoprene emissions are quite uniform in all wind directions.

After having comments related to this topic from Referee #1 as well, we decided to delete most of the isoprene analysis because the more detailed research would be impossible. However, the basic analysis was left into the manuscript, but we pointed out that the values are not representative for further use (e.g. for models).

**page 8 line 13: In Figure 3 it can be seen that acetonitrile and acetone seem to be emitted from the built sector. Could those be the result of solvent use at the**

**University buildings?**

Acetone is most probably coming from University buildings, especially from Chemistry Department. This is shortly discussed in page 13, lines 16–19 (original manuscript). Acetonitrile could also originate from similar sources and we added this speculation into the text (Section 3.2.3).

**page 8 line 26: The annual trend in the concentration of the aromatics will also strongly depend on the boundary layer height. Atmospheric background mixing ratios of benzene are much higher in winter in the northern hemisphere with over 100ppt, but local enhancements in an urban area are probably more driven by the boundary layer height than lifetimes. Again it would be very helpful to look at annual and diurnal cycles of mixing ratios in detail as well.**

This is true and local sources affect on diurnal trends of concentrations. Boundary layer heights do also have an effect and we discuss more about the topic in the revised manuscript (Section 3.1). We added also more discussion about annual and diurnal trends.

**page 9 line 19-23: I think it is problematic to compare CO/CO2 ratios with other studies without taking the strong decreasing trend of CO into account. Over the past decade(s) CO and VOCs have decreased by several percent every year. The discussion should take this trend into account. Also, there are other sources of CO in a city compared to cars driving in a tunnel, e.g cold starts, (as mentioned in the text), domestic burning and other residential and commercial combustion sources. I would therefore delete the comparison with the tunnel study and look at other papers that show CO/CO2 enhancement ratios.**

As suggested, we deleted the tunnel study section. Besides, a study from Harrison et al. (2012) was compared with our results. We also mention that CO emissions from traffic have had a decreasing temporal trend which may explain partly the differences between our study and Famulari et al. (2010). See also the response for Referee #1.

**page 9 lines 24-31: I agree that cold starts are likely an important source of CO and VOCs in the built sector, but I am wondering if the high CO and aromatics emissions in the afternoon could also be explained by domestic burning. Acetonitrile is generally used as a tracer for biomass burning, but it is not a good tracer for domestic burning, because N emissions are generally smaller from wood than foliage burning (e.g. Yokelson et al ACP 2014). So the lack of acetonitrile fluxes by itself is not a reason to discard domestic burning as a major source of CO and VOCs in winter. The domestic burning should have a strong annual cycle. I am not sure, if without the annual cycle measurement of CO, there is enough evidence to look for this source here.**

This might be a good explanation and was added to the text (Section 3.2). Unfortunately, CO-fluxes were measured during two months only, thus, studying annual cycles was not possible. Domestic burning near the site is probably more related to warming Saunas and using fireplaces. Most of the houses in Helsinki and within the flux source area are warmed by district heating, thus, these emissions from residences do not necessarily have the annual cycle.

**page 10 line 9: Why did you choose Sep-May and not Dec-Feb? Shouldn't that give you a better contrast?**

The original plan was to divide the data to two classes due to a better statistics. However, as both Referees suggested, we divided the data into four classes according to the seasons to have better contrast between colder and warmer periods. The text and some results (e.g. related to biogenic contribution) were modified accordingly. See also the response for Referee #1.

**page 10 line 15: I don't know about the fuels in Finland, but methanol, acetaldehyde and acetone are usually not ingredients of gasoline. In many places, especially the USA and Brazil, gasoline contains a lot of ethanol, but usually no other oxygenated VOCs.**

[Figure]

In Finland, a popular 95E10 contains ethanol ($< 10\%$) and methanol (< 3%). Of course, this does not mean necessarily traffic related methanol emissions but they might be possible. According to Caplain et al. (2006), acetone and acetaldehyde have also tail pipe emissions.

We added more discussion about the topic to the revised manuscript (Section 3.2.1).

**page 10 lines 25-28: It is probably correct that in the road sector most aromatics are from traffic, but in general toluene and to a lesser extend C2-aromatics have large non traffic related sources such as solvents, paints and paint thinners. This should be mentioned here.**

We agree with this statement and added discussion about additional sources to the text. We believe that those additional sources should have only a relatively minor effect on fluxes in the road sector but of course they might be more important in the built sector. See also the comment from Editor and our response.

**page 11 line 9: The total traffic related flux of aromatics is calculated here and with this the fraction of the traffic to total aromatics flux can be estimated. As I mentioned earlier, this would be a very important result. Is the error of 1.2+/-0.2 g/m2/yr correct? Looking at the error estimate in the equation two lines above, this seems low?**

The error analysis was based on the uncertainty estimates of the slope ($29 \pm 5$...) between the fluxes and the traffic rates, i.e. we assumed that the intercept described non-traffic sources and its uncertainty estimate was excluded from the calculations. The error estimate is correct but on the other hand, it excludes possible systematic errors, such as calibration or possible errors of the traffic counts. We clarified this in the text.

**page 11 line 15: This is the only time Figure 9 is mentioned in this section of the text and is only briefly mentioned later on. Either this figure needs to be**

**explained better or deleted.**

We agree. We simplified the figure and added more discussion about it into the text (Section 3.1).

**page 11 lines 20-28: It is clear that even in the road sector other sources besides traffic are strongly contributing to the VOC emissions, which can be seen in all the low correlation coefficients given in this paragraph. This would be a good place to mention the difficulty in the source apportionment again.**

This is true and we expanded the discussion about the difficulties in the revised manuscript (Section 3.2.1). However, low correlations do not necessarily mean that also other sources than traffic have major contribution on – for example – aromatic fluxes. Flux measurements were just quite noisy for many compounds, decreasing also correlations.

**page 11 lines 29-34: Here I am wondering again, if cycloalkanes are contributing to the signal on mass 69.**

This is totally possible, at least cyclohexane might contribute to the measured concentrations at m/z 69. We are not sure whether cycloalkanes contribute also to the measured flux at m/z 69 or not. Nevertheless, we mention cycloalkanes in the text and Table 2 in the revised manuscript.

**page 12 line 21: The isoprene emissions, at least for the road and vegetation sectors, are of the same magnitude, but the CO2 emissions look very different. There is a very clear signature of CO2 uptake in the vegetation sector, and therefore one would expect to have similar CO2 uptake in the road sector and as a result the anthropogenic CO2 flux is underestimated. Is this effect taken into account in this manuscript? Can this be used to estimate the flux and be compared to the Jaervi et al 2012 paper?**

Carbon dioxide emissions from the vegetation sector are relatively low, thus, the net

carbon uptake becomes visible during summer. In the road sector, this kind of behaviour cannot be seen as the $CO_2$ emissions from traffic dominate the net exchange during all seasons.

Järvi et al. (2012) took the possible $CO_2$ uptake to account by using $CO_2$ flux data from snow covering season only. However, in this study – for example in Fig. 6 – the $CO_2$ uptake was not taking to account as it would be very tricky. However, the possible uptake by the vegetation is mentioned in the revised manuscript.

**page 13 lines 14-15 and page 13 lines 26-27: It is not clear to me, how these contributions are actually calculated, the biogenic monoterpene contribution and the non-traffic related contributions of anthropogenic VOCs. As I mentioned earlier, this is one of the more important results and it should be explained in detail how these contributions or "source attribution" is calculated.**

In the original manuscript, the average fluxes from September–May were compared with the average fluxes from June–August. However, as we present seasonal cycles instead of two periods only in the revised manuscript, that comparison would not make sense anymore. Therefore, we decided to determine the biogenic contribution by comparing the fluxes in the range of $T < 10°$C with the average fluxes from summer. We could have also compared the average winter fluxes with the average summer fluxes. However, the statistics was better with the temperature criteria.

The new procedure increased the biogenic contribution (for methanol: 25% → 40%) but the estimate should be now more accurate and also more clear for the reader. See also the responses for Referee #1.

**page 14 lines 1-8: I agree that acetonitrile from the built sector is likely from solvent use in the chemistry buildings, but again domestic burning cannot be well characterized using acetonitrile due to the low emission rate from domestic burning.**

This is a good statement and was added to the text (Section 3.2.3).

**Figure 2: please label the sectors and add typical footprints.**

We labelled the sectors and added the cumulative 80% footprint.

**Figure 3, 6 and 9: a label would make the figures much easier to look at.**

Labels were added to Figs. 3, 6 and 9 (Figs. 3, 4 and 9 in the revised manuscript).

**Figure 7: It would be worth showing the other sectors for comparison.**

We added CO-CO$_2$ comparisons from other sectors as well (Fig. 9). However, comparison to traffic counts is possible for road sector only because in other directions the nearest roads were more far away and traffic counts were not counted for those directions.

**Technical Comment: There are so many small grammatical errors everywhere in the manuscript, mainly missing articles and prepositions, that I can not list them all here. I would suggest an additional proofreading.**

The revised manuscript was carefully proofread.

References

Caplain, I., Cazier, F., Nouali, H., Mercier, A., Déchaux, J.-C., Nollet, V., Joumard, R., André, J.-M., and Vidon, R.: Emissions of unregulated pollutants from European gasoline and diesel passenger cars, Atmospheric Environment, 40, 5954–5966, 2006.

Famulari, D., Nemitz, E., Marco, C. D., Phillips, G. J., Thomas, R., House, E., and Fowler, D.: Eddy-covariance measurements of nitrous oxide fluxes above a city, Agricultural and Forest Meteorology, 150, 786–793, 2010.

Hellén, H., Hakola, H., and Laurila, T.: Determination of source contributions of NMHCs in Helsinki (60°N, 25°E) using chemical mass balance and the Unmix multivariate receptor models, Atmospheric Environment, 37, 1413–1424, 2003.

Hellén, H., Hakola, H., Pirjola, L., Laurila, T., and Pystynen, K.-H.: Ambient air concentrations, source profiles, and source apportionment of 71 different C2-C10 volatile organic compounds in urban and residential areas of Finland, Environmental science & technology, 40, 103–108, 2006.

Hellén, H., Tykkä, T., and Hakola, H.: Importance of monoterpenes and isoprene in urban air in northern Europe, Atmospheric Environment, 59, 59–66, 2012.

Järvi, L., Nordbo, A., Junninen, H., Riikonen, A., Moilanen, J., Nikinmaa, E., and Vesala, T.: Seasonal and annual variation of carbon dioxide surface fluxes in Helsinki, Finland, in 2006–2010, Atmospheric Chemistry and Physics, 12, 8475–8489, 2012.

Järvi, L., Nordbo, A., Rannik, ü., Haapanala, S., Riikonen, A., Mammarella, I., Pihlatie, M., and Vesala, T.: Urban nitrous-oxide fluxes measured using the eddy-covariance technique in Helsinki, Finland, Boreal Environment Research, 19B, 108-121, 2014.

Vesala, T., Järvi, L., Launiainen, S., Sogachev, A., Rannik, ü., Mammarella, I., Siivola, E., Keronen, P., Rinne, J., Riikonen, A., and Nikinmaa, E.: Surface–atmosphere interactions over complex urban terrain in Helsinki, Finland, Tellus B, 60, 188–199, 2008.

---

## Author Comment (AC2) · 31 May 2016

We thank the editor for the helpful comments and suggestions. Editor comments are bolded whereas author responses are written in a normal text.

**Looking at the map provided in the figure 2, the site seems to be particularly influenced by green vegetation. Even the sector identified as buildings is comprised of a fairly large fraction of vegetation, rather unique compared to the urban UK sites the authors compare their measurements with. Consistently this sector does not show a significantly different pattern of m/z 69 emissions. Perhaps a better description of this sector would be to call it 'urban residential sector with vegetation'.**

[Figure]

This is true, the sectors have only small differences in vegetation coverage (Table 1). Therefore, the names of the sectors are somewhat vague but have been widely used in previous publications concerning the same site (Vesala et al., 2008; Järvi et al., 2012, 2014). In that sense, we would still like to use the original names. However, we expanded the discussion about the land use (section 2.1).

Temperature and PPFD normalized isoprene emissions were around the same from all wind directions. However, absolute values differ because the ambient temperature was typically the lowest when the wind blew from the built sector.

**BTEX emissions: The sector identified as road boarders to what it seems like an industrial complex (for example: at a distance of about 300-400 m a smoke stack is evident on google earth). It is argued that this sector is primarily influenced by road traffic. The influence of additional BTEX sources in this sector (other than traffic) could perhaps be obtained by explicitly comparing toluene to benzene fluxes during rush hour peaks with other periods. The upper limit of traffic related emission ratios should be close to 2 (1.9) based on the emission factor database for the average European fleet. The authors compare their measurements to other cities. In this context it is noted that Mexico City seems to be a special place with respect to many of the measured VOC fluxes. For example toluene measurements by Velasco et al., 2009, were thought to be influenced by local application of resin surrounding the flux tower resulting in toluene / benzene flux ratios of about 8-10. Measurements by Karl et al., 2009, reported a city wide average ratio of about 3.2 for Mexico City and concluded that about 60-70% of toluene could be due to evaporative emissions. Figure 11: It is noted that a correlation of fluxes between some compounds (such as toluene) and CO2 needs to be discussed with caution. For example most of traffic related toluene emissions are evaporative and not produced by the ICE - thus not intrinsically linked to CO2 tailpipe emissions. This is fundamentally different for benzene emissions for example, which are much more closely related to tail-pipe emis-**

**sions.**

We thank for the good suggestion. We calculated the ratio between the average toluene flux and the average benzene flux for two cases: all the data and periods when the traffic rate was over 2000 vehicles per hour. The ratios were $2.9 \pm 0.7$ and $3.1 \pm 1.0$, respectively, indicating non-traffic related toluene sources. Interestingly, the ratio was higher for the high traffic period but the difference was statistically insignificant. On the other hand, the toluene fluxes alone followed also well the traffic counts with an intercept of $4 \pm 5$ ng m$^{-2}$s$^{-1}$. Of course, toluene might still have evaporative emissions, but the traffic related emissions seemed to be still more important source if the offset is assumed to describe the non-traffic related emissions. We expanded the discussion related to non-traffic related sources of toluene and C$_2$-benzenes (sections 3.2.1 and 3.3). We also added Table A2 to the manuscript which contains the VOC to benzene flux ratios for each season.

We agree that the terrain in quite heterogeneous consisting of different land covers, but the old ceramic factory to which the editor likely refers to is not active anymore. The nearest industrial and workshop activities are over 800 -1000 m to the east south but these start to be already outside the flux footprint which typically expands less than that (e.g. Ripamonti et al. 2013, Figure 2). Also these emissions sources did not show up the in the Fig. S2 where the average VOC fluxes are plotted as a function of wind direction.

We also agree that the VOC fluxes are not totally comparable with the CO$_2$ fluxes as many VOCs have also other anthropogenic sources than CO$_2$ does. We pointed this out more carefully in the revised manuscript (section 3.3). The message of the comparison was to show that the low VOC fluxes measured in Helsinki are rather sensible when taking to account also low CO$_2$ fluxes, indicating less intense anthropogenic activities, such as traffic related emissions. However, we did not want to argue that those two emissions should have necessarily a linear dependency.

[Figure]

We included also Karl et al. (2009) to the discussion. We admit that the study by Velasco et al. (2009) was done at a unique location, but that was one of the rare studies which provided both the $CO_2$ and VOC fluxes from the same urban location.

**References**

Järvi, L., Nordbo, A., Junninen, H., Riikonen, A., Moilanen, J., Nikinmaa, E., and Vesala, T.: Seasonal and annual variation of carbon dioxide surface fluxes in Helsinki, Finland, in 2006–2010, Atmospheric Chemistry and Physics, 12, 8475–8489, 2012.

Järvi, L., Nordbo, A., Rannik, Ü., Haapanala, S., Riikonen, A., Mammarella, I., Pihlatie, M., and Vesala, T.: Urban nitrous-oxide fluxes measured using the eddy-covariance technique in Helsinki, Finland, Boreal Environment Research, 19B, 108–121, 2014.

Karl, T., Apel, E., Hodzic, A., Riemer, D. D., Blake, D. R., and Wiedinmyer, C.: Emissions of volatile organic compounds inferred from airborne flux measurements over a megacity, Atmospheric Chemistry and Physics, 9, 271–285, 2009.

Ripamonti G., Järvi L., Molgaard, M., Hussein T., Nordbo A. and Hämeri K. (2013). The effect of local sources on aerosol particle number size distribution, concentrations and fluxes in Helsinki, Finland. Tellus B 65.

Vesala, T., Järvi, L., Launiainen, S., Sogachev, A., Rannik, Ü., Mammarella, I., Siivola, E., Keronen, P., Rinne, J., Riikonen, A., and Nikinmaa, E.: Surface-atmosphere interactions over complex urban terrain in Helsinki, Finland, Tellus B, 60, 188–199, 2008.